# Integrative proteomics reveals principles of dynamic phosphosignaling networks in human erythropoiesis

Özge Karayel[1,†] (ID), Peng Xu[2,†], Isabell Bludau[1], Senthil Velan Bhoopalan[2] (ID), Yu Yao[2], Ana Rita Freitas Colaco[3], Alberto Santos[3], Brenda A Schulman[4,*] (ID), Arno F Alpi[4,**] (ID), Mitchell J Weiss[2,***] (ID) & Matthias Mann[1,3,****] (ID)

## Abstract

Human erythropoiesis is an exquisitely controlled multistep developmental process, and its dysregulation leads to numerous human diseases. Transcriptome and epigenome studies provided insights into system-wide regulation, but we currently lack a global mechanistic view on the dynamics of proteome and post-translational regulation coordinating erythroid maturation. We established a mass spectrometry (MS)-based proteomics workflow to quantify and dynamically track 7,400 proteins and 27,000 phosphorylation sites of five distinct maturation stages of *in vitro* reconstituted erythropoiesis of CD34[+] HSPCs. Our data reveal developmental regulation through drastic proteome remodeling across stages of erythroid maturation encompassing most protein classes. This includes various orchestrated changes in solute carriers indicating adjustments to altered metabolic requirements. To define the distinct proteome of each maturation stage, we developed a computational deconvolution approach which revealed stage-specific marker proteins. The dynamic phosphoproteomes combined with a kinome-targeted CRISPR/Cas9 screen uncovered coordinated networks of erythropoietic kinases and pinpointed downregulation of c-Kit/MAPK signaling axis as key driver of maturation. Our system-wide view establishes the functional dynamic of complex phosphosignaling networks and regulation through proteome remodeling in erythropoiesis.

**Keywords** (Phospho)proteomics; CRISPR/Cas9 library screen; human erythropoiesis; SLC; systems biology
**Subject Categories** Development; Proteomics
**Mol Syst Biol. (2020) 16: e9813**

## Introduction

Human erythropoiesis is a multistep developmental process that maintains stable erythroid homeostasis throughout life and replenishes more than 200 billion erythrocytes lost by senescence in healthy humans (Palis, 2014). Lineage-committed erythroid progenitors, including burst-forming unit-erythroid (BFU-E) and their colony-forming unit-erythroid (CFU-E) progeny, undergo enormous expansion, followed by morphological signs of terminal maturation. The first recognizable erythroid precursors are proerythroblasts (ProE), which mature progressively into early basophilic (EBaso) and late basophilic (LBaso) erythroblasts, polychromatic erythroblasts (Poly), orthochromatic (Ortho) erythroblasts, and reticulocytes. Terminal erythroid maturation is distinguished by progressive reductions in proliferative capacity and cell size, chromatin condensation, loss of most organelles including the nucleus, and remarkable streamlining of the proteome with expression of specialized cytoskeletal and plasma membrane proteins and finally massive accumulation of hemoglobin (Zhao, Yang *et al*, 2016; Moras, Lefevre *et al*, 2017; Nguyen, Prado *et al*, 2017). This finely tuned developmental process generates mature erythrocytes with the highly specialized function of circulatory oxygen/carbon dioxide transport.

Our knowledge of human erythropoiesis has been greatly advanced by *in vitro* differentiation systems in which primary multipotent CD34[+] hematopoietic stem/progenitor cells (HSPCs) are cultured with defined cytokines and other bioactive components to generate reticulocytes (Seo, Shin *et al*, 2019). Erythropoiesis is controlled by the essential cytokines, stem cell factor (SCF) and erythropoietin (EPO), and their cognate receptors, c-Kit and EPOR (Nocka, Majumder *et al*, 1989; Wu, Klingmuller *et al*, 1995; Broudy, 1997; Wu, Klingmuller *et al*, 1997; Zhang & Lodish, 2008; Ingley,

1  Department of Proteomics and Signal Transduction, Max Planck Institute of Biochemistry, Martinsried, Germany
2  Department of Hematology, St. Jude Children's Research Hospital, Memphis, TN, USA
3  Novo Nordisk Foundation Center for Protein Research, Faculty of Health Sciences, University of Copenhagen, Copenhagen, Denmark
4  Department of Molecular Machines and Signaling, Max Planck Institute of Biochemistry, Martinsried, Germany
   *Corresponding author. Tel: +49 89 8578 2472; E-mail: schulman@biochem.mpg.de
   **Corresponding author. Tel: +49 89 8578 2480; E-mail: aalpi@biochem.mpg.de
   ***Corresponding author. Tel: +1 901 595 3760; E-mail: mitch.weiss@stjude.org
   ****Corresponding author. Tel: +49 89 8578 2557; E-mail: mmann@biochem.mpg.de
   †These authors contributed equally to this work

2012). In general, c-Kit acts to promote progenitor proliferation during early erythropoiesis, while EPOR fosters survival and maturation at later stages, although there is substantial overlap in their activities and some evidence for cross-regulation (Klingmuller, 1997; Wu *et al*, 1997; Wojchowski, Gregory *et al*, 1999). Moreover, c-Kit and EPOR trigger remarkably similar signaling pathways including Ras/Raf/MAPK, PI3K/Akt, and JAK2/STAT5 (Carroll, Spivak *et al*, 1991; Miura, Miura *et al*, 1994; Linnekin, Mou *et al*, 1997; Socolovsky, Fallon *et al*, 1999; Bouscary, Pene *et al*, 2003; Wandzioch, Edling *et al*, 2004; Ghaffari, Kitidis *et al*, 2006). In concert with cytokine signaling, several key erythroid-restricted transcription factors (including GATA-1, FOG-1, SCL/TAL-1, EKLF/KLF1) associate with generalized cofactors to activate the transcription of erythroid-specific genes and suppress those of alternate lineages (Pevny, Simon *et al*, 1991; Perkins, Sharpe *et al*, 1995; Shivdasani, Mayer *et al*, 1995; Cross & Enver, 1997; Tsang, Visvader *et al*, 1997; Cantor & Orkin, 2002; Akashi, He *et al*, 2003; Hattangadi, Wong *et al*, 2011).

While focused studies on erythroid cytokine signaling and transcription factors have generated tremendous functional insights into erythropoiesis, they do not provide a system-wide view. A comprehensive view of erythroid gene expression has been provided by global analysis of erythroid transcriptomes and the epigenome in purified bulk populations and single cells ((Tusi, Wolock *et al*, 2018) and reviewed in (An, Schulz *et al*, 2015)). These approaches necessarily use global mRNA levels as proxies of protein abundance and infer signaling activity indirectly. A truly system-wide understanding of post-transcriptional and translational mechanisms that drive and coordinate terminal maturation is clearly still lacking. Such a dynamic map would complement transcriptome studies to broadly describe the molecular basis of the pathways involved and to understand how cytokine receptor signaling and transcription factors together shape the erythroid proteome.

In contrast to transcriptome and epigenetic studies of erythropoiesis, relatively few proteome studies to date provide a global analysis of the protein landscape. Due to technical limitations, these studies examined only selected maturation stages in limited depth or focused on defined protein families (Pasini, Kirkegaard *et al*, 2006; Roux-Dalvai, Gonzalez de Peredo *et al*, 2008; Bell, Satchwell *et al*, 2013; Gautier, Ducamp *et al*, 2016; Wilson, Trakarnsanga *et al*, 2016; Liu, Zhang *et al*, 2017; Amon, Meier-Abt *et al*, 2019; Gillespie, Palii *et al*, 2020). A recent analysis described dynamic changes in protein expression during *in vitro* erythroid differentiation of CD34[+] HSPCs (Gautier *et al*, 2016). However, because relatively large numbers of cells were required for proteomic characterization, this study examined semisynchronous erythroid cultures consisting of cells at different stages of maturation. This study has employed data-dependent acquisition (DDA) method, which selects and subsequently fragments the most intense peptides into product ions that are analyzed in the mass analyzer, combined with label-free quantification. Recently, data-independent acquisition (DIA) has emerged as a powerful alternative to DDA for discovery-oriented, unbiased proteomics analysis (Gillet, Navarro *et al*, 2012; Bruderer, Bernhardt *et al*, 2015; Kelstrup, Bekker-Jensen *et al*, 2018; Ludwig, Gillet *et al*, 2018). In contrast to the semi-stochastic nature of DDA, DIA generates comprehensive product ion maps by fragmenting all co-eluting peptide ions within predefined mass-to-charge ($m/z$) windows and acquiring them together (Venable, Dong

*et al*, 2004). It results in more accurate quantification with fewer missing values across conditions and higher identification rates over a larger dynamic range. Given the superior performance of DIA for sensitive and reproducible MS measurements and the advances in the sample preparation methods for high sensitivity phosphoproteomics (Aebersold & Mann, 2016; Bekker-Jensen, Kelstrup *et al*, 2017), we reasoned that it may now be possible to obtain accurate high coverage proteome and phoshoproteome quantification from relatively low numbers of purified erythroid precursors at distinct developmental stages.

We developed a pipeline combining fluorescence activated cell sorting (FACS) enrichment procedures with our state-of-the-art proteomics workflow. We uncovered the temporal staging of developmental regulation through proteome remodeling. To identify the distinct proteome defining each maturation stage from proerythroblast to orthochromatic erythroblast, we developed a bioinformatic deconvolution approach which revealed stage-specific proteins and protein families. Importantly, our proteomics workflow enabled detection of more than one thousand membrane proteins and identified distinct combinations of solute carrier (SLC) family proteins as stage-specific maturation markers. Pursuing post-translational regulation further, in-depth sensitive quantitation of the global phosphoproteome with our EasyPhos platform (Humphrey, Azimifar *et al*, 2015; Humphrey, Karayel *et al*, 2018) provided direct evidence for intricate developmental stage-specific regulation by post-translational modification. To functionally explore the identified signaling modules, we performed a kinome-targeting CRISPR/Cas9 screen, which in combination with our proteomic studies, identified distinct signaling requirements for erythroid maturation. Focusing on networks among over 27,000 phosphosites and kinase functions uncovered the sequential attenuation of c-Kit and EPOR/JAK2 signaling, pinpointing downregulation of Ras/MAPK signaling in promoting terminal maturation. Our system-wide data provide a wealth of molecular information regarding the functional dynamics of complex phosphosignaling networks in erythropoiesis, expanding our knowledge and data for cellular principles of regulation through proteome remodeling.

## Results

### Establishing stage-specific proteomes of human erythropoiesis

To investigate the remodeling of the proteomics landscape during human erythropoiesis, we cultured human peripheral blood-derived CD34[+] HSPCs under conditions to support erythroid differentiation (Materials and Methods). We obtained highly enriched populations of erythroid precursors at specific developmental stages by FACS using CD235a (GYPA), CD49d (ITGA4), and Band 3 (SLC4A1) markers (Figs 1A and B, and EV1A) (Hu, Liu *et al*, 2013). We isolated early maturation stages (progenitors, ProE, EBaso, LBaso) after 7 days of culture, while later maturation stages (LBaso, Poly, and Ortho) were purified at day 14. LBaso stage precursors were isolated after both 7 days and 14 days of culture using the same markers (Fig 1B). Note that SCF was present at 7 days but not at 14 days. Purified cell populations were morphologically homogeneous as judged by May–Grünwald–Giemsa staining (Fig EV1B). Due to relatively low cell yields, ProE and EBaso populations were combined

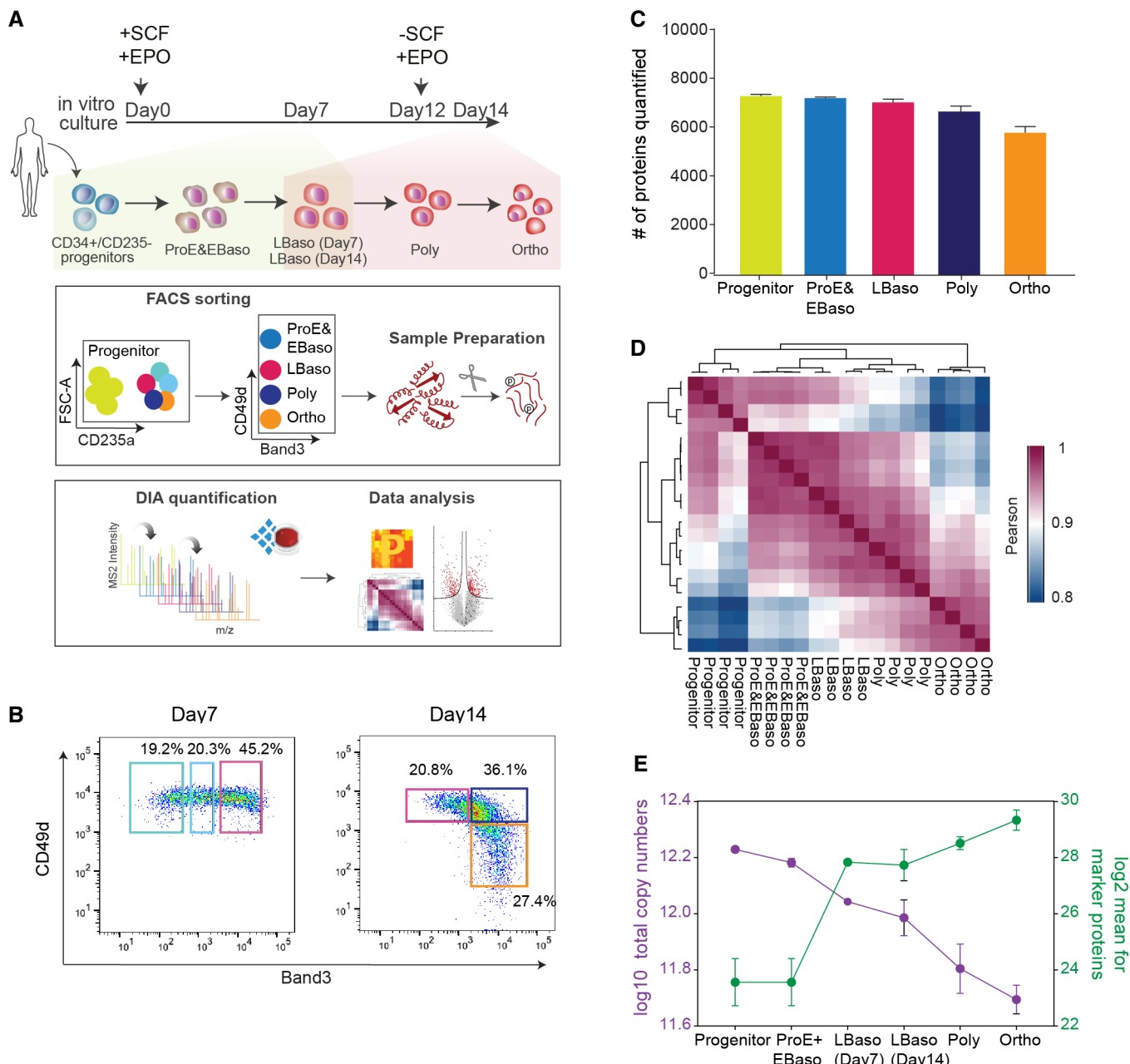

**Figure 1. Establishing differentiation stage-specific proteomes of human erythropoiesis.**

A   Top panel depicts culture conditions for *in vitro* erythroid differentiation of CD34[+] cells. Shading indicates the presence of SCF and EPO (yellow), or EPO alone (pink). Note that SCF was present at 7 days but not at 14 days. The lower panels indicate the workflow of our study, including FACS gating/sorting strategy of erythroid precursors and single shot DIA analysis.

B   FACS gating regime to enrich for ProE, EBaso, LBaso, Poly, and Ortho erythroblasts.

C   Number of different proteins quantified in each differentiation stage. The mean values ± SD of four biological replicates are shown.

D   Correlation based clustering illustrating the reproducibility between biological replicates. High (1.0) and lower (0.8) Pearson correlations are denoted in pink and blue, respectively.

E   Estimated copy numbers of total molecules (purple) and mean copy numbers of the proteins with GO annotations "erythrocyte maturation" and "heme biosynthesis" (green) per cell across maturation stages. The mean values ± SEM of four biological replicates are shown.

in equal cell numbers prior to subsequent analysis. The resulting five populations/stages are henceforth color-coded as follows: progenitors (mostly CFU-E) (Hu *et al*, 2013; Li, Hale *et al*, 2014; Yan, Wang *et al*, 2017), yellow; ProE/EBaso, blue; LBaso day 7,

light pink; LBaso day 14, dark pink; Poly, dark blue; and Ortho, orange (Fig 1A and B).

Each population was processed in four biological replicates, and their tryptic peptides were analyzed in single shots in

data-independent acquisition (DIA) mode (Materials and Methods, Fig 1A). To generate a project-specific spectral library necessary for the DIA approach, we separated peptides by high-pH reversed-phase chromatography into fractions, followed by data-dependent acquisition (DDA) and analysis with Spectronaut. The resultant library contained more than 9,000 protein groups, 7,479 of which could be matched into the DIA runs of at least one maturation stage (*q*-value < 1% at protein and precursor levels, Fig 1C). In the DIA method, small *m/z* precursor windows are fragmented in a cyclical manner, which turned out to be crucial for preserving the dynamic range of peptide detection in the presence of the very large hemoglobin peptide peaks that would otherwise complicate analyses at later developmental stages. Remarkably, 84% of all detected proteins were consistently quantified at varying levels across all maturation stages and a relatively small percentage was only matched in a single stage. Quantitative accuracy was high, with Pearson correlations > 0.95 and CVs < 20% for 72% of all proteins between the four biological replicates (Figs 1D and EV1C). MS signals spanned abundance ranges of five (progenitors) to seven (Ortho) orders of magnitude. As expected, globin proteins increased by approximately one thousand-fold from progenitor to Ortho stage (Fig EV1D).

As biological interpretation is facilitated by absolute rather than relative concentration measurements, we employed the "proteomic ruler" method, which uses the fixed relationship between histones and DNA to estimate proteome-wide copy numbers per cell (Wisniewski, Hein *et al*, 2014). Considering that chromatin condensation during erythropoiesis is associated with partial release of major histones from the nucleus and subsequent degradation in the cytoplasm (Zhao, Mei *et al*, 2016), we first assessed the overall histone content of cells in our system, which indeed declined with progressive differentiation (Fig EV1E). Taking this into account for the proteomic ruler calculations, we measured an almost four-fold reduction in total protein copy numbers per cell during differentiation with median copy numbers dropping from 23,380 ± 371 in progenitors to 12,395 ± 1,342 at the LBaso stage (Figs 1E and EV1E and Dataset EV1). In contrast, the average copy numbers of proteins annotated as "erythrocyte maturation" and "heme biosynthesis" by Gene Ontology (GO) increased by approximately 50-fold from progenitor to Ortho stage (Fig 1E and Dataset EV1). Quantitative comparison and copy number estimation of LBaso stages isolated at either day 7 or day 14 confirmed their close resemblance at the global proteome level, including marker proteins, such as GYPA, CD49d, Band 3, c-Kit, and several hemoglobin subunits that did not significantly change (Figs 1E and EV2A–C). Thus, they were combined for further proteomic analysis unless otherwise noted.

## Dynamic and stage-specific proteome remodeling in erythropoiesis

The five stages of human erythropoiesis clustered separately by principal component analysis (PCA) with very high concordance between replicates (Fig 2A). Hierarchical clustering of 4,316 proteins with statistically different expressions (ANOVA, FDR < 0.01) revealed drastic differences in the stage-specific proteomes. Rather than straightforward increase or decrease in protein levels across differentiation, proteins cluster into one of six distinct profiles of temporal co-expression dynamics (Fig 2B and Dataset EV2).

In addition to known developmental themes in each cluster, GO enriched terms point to novel state-specific regulation (summarized in Fig EV3). In pairwise comparisons between successive stages, 2,157 proteins (29%) changed significantly at the first transition (Fig 2C). The overall proteome was more stable from ProE/EBaso to Poly stages, with 8.5% proteins up- or downregulated. In contrast, almost 20% of the proteome significantly changed in the last investigated transition, reflecting the specialization toward mature erythrocytes (Fig 2C).

To discover unique stage-specific marker proteins, we compared all stages against each other (Fig EV4A). Interestingly, the Poly stage can be distinguished by the centralspindlin and chromosomal passenger complexes (Benjamini–Hochberg, FDR < 0.01). These proteins regulate cytokinesis in the late stages of cell division and also likely participate in erythroblast enucleation. Indeed, mutations in the kinesin KIF23B cause congenital dyserythropoietic anemia associated with erythroid multinuclearity and impaired erythropoiesis (Liljeholm, Irvine *et al*, 2013). This analysis, like the ANOVA results, revealed the most drastic proteome changes occurring at the transition from progenitors to ProE/EBaso and from Poly to Ortho (Fig EV4A). The cumulative proteome remodeling from the progenitors was reflected in a very large fraction of differentially represented proteins at the later maturation stages, Poly and Ortho (44 and 57%, respectively, two sample test, FDR < 0.01 and S0 = 0.1) (Fig 2D).

Taken together, our stage-specific proteomic data enable accurate, quantitative, and in-depth monitoring of global protein expression during human erythropoiesis. The identified proteins are potentially important for the functional specialization of erythroid cells toward mature erythrocytes and represent excellent starting points for more detailed mechanistic studies.

## Dramatic remodeling of the transmembrane proteome in erythropoiesis

Our data capture distinct regulation of proteins that contribute to the highly specialized erythroblast membrane at later developmental stages. Despite identification of several transmembrane proteins as markers of erythropoiesis over the years (Chen, Liu *et al*, 2009), there is still limited system-wide information on them. The use of sodium deoxycholate as an ionic detergent optimized our lysis and digestion protocol to efficiently denature and solubilize hydrophobic proteins. This enabled unbiased access to the membrane-associated proteome and provided a comprehensive view of membrane proteins during differentiation. Across the differentiation stages, we quantified 1,033 plasma membrane proteins (~21% of the total genome-encoded plasma membrane proteome in humans and ~14% in our study), of which 692 changed significantly (Dataset EV1). Our data identified a plethora of new examples that will aid in pinpointing maturation stages and in better understanding of erythroid biology.

Of the significantly changing membrane proteins, we could map 86% to pathways, with transport of small molecules across plasma membranes among the most represented (p 8.7 E-09). Further functional classification revealed involvement of several known protein families, including the ATP-binding cassette (ABC) transporters, and showed markedly strong enrichment of "SLC (solute carrier)-mediated transmembrane transport", confirming previous transcriptomic

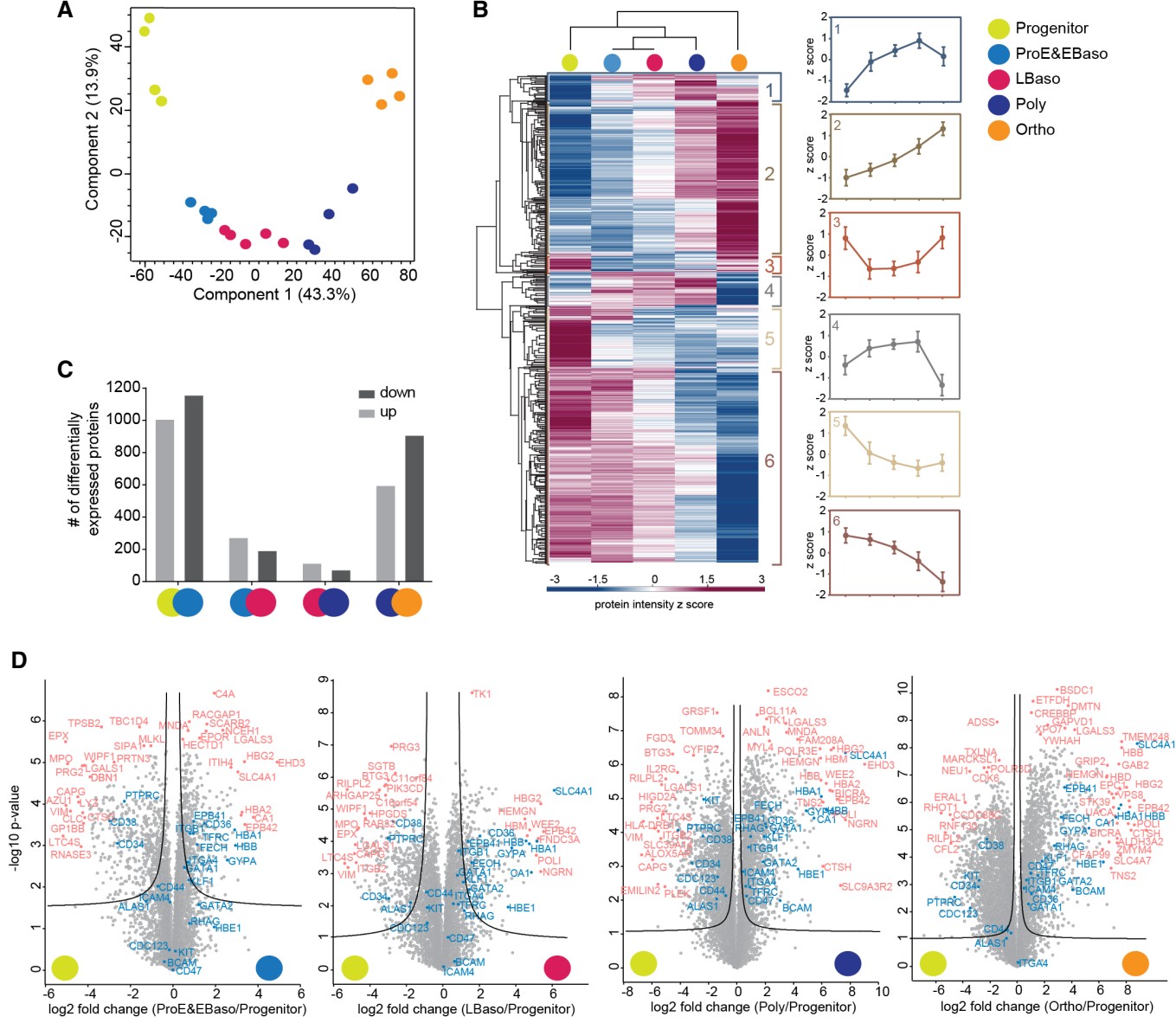

**Figure 2. Dynamic and stage-specific proteome remodeling in erythropoiesis.**

A   PCA of differentiation stages along with their biological replicates based on their proteomic expression profiles.
B   Heat map of z-scored protein abundances (log2 DIA intensities) of the differentially expressed proteins (ANOVA, FDR < 0.01) after hierarchical clustering reveals six main profiles. Mean z-scores with standard errors (SEM) of four biological replicates are shown in each stage.
C   Number of differentially expressed proteins in pairwise comparisons of successive stages of human erythroid differentiation.
D   Individual volcano plots of the (-log10) *P*-values versus the log2 protein abundance differences between progenitor and the four differentiation stages. Selected significant proteins and previously reported marker proteins are labeled in pink and blue, respectively and significance lines (FDR < 0.01) are shown.

analyses (Hu *et al*, 2013; An, Schulz *et al*, 2014) (Figs 3A and EV4B). The roles of SLCs in biology have arguably been understudied, but now there are systematic efforts characterizing their roles (Cesar-Razquin, Snijder *et al*, 2015). Notably, since identification of "Band 3" as a solute carrier protein (SLC4A1) 35 years ago (Kopito & Lodish, 1985), it has become clear that SLC proteins must have widespread roles in erythropoiesis. Remarkably, our data quantified 101 SLCs, 68 of which significantly change in at least one transition (Fig 3B), likely reflecting remarkable changes in metabolic requirements along the stages of maturation. As summarized in Table EV1,

62 of these have known or purported substrates associated with them.

Only 22 of the significantly regulated SLCs have previously been linked to erythrocytes, erythropoiesis, or anemia. For instance, Mitoferrin-1 (SLC25A37), with a continuous upregulation during erythroid maturation, is a mitochondrial iron importer essential for heme biosynthesis in erythroblasts (Shaw, Cope *et al*, 2006). For some SLCs, roles in transporting nutrients including glucose and amino acids, and ions such as zinc, and necessary functions as redox regulators in erythropoiesis have already been described

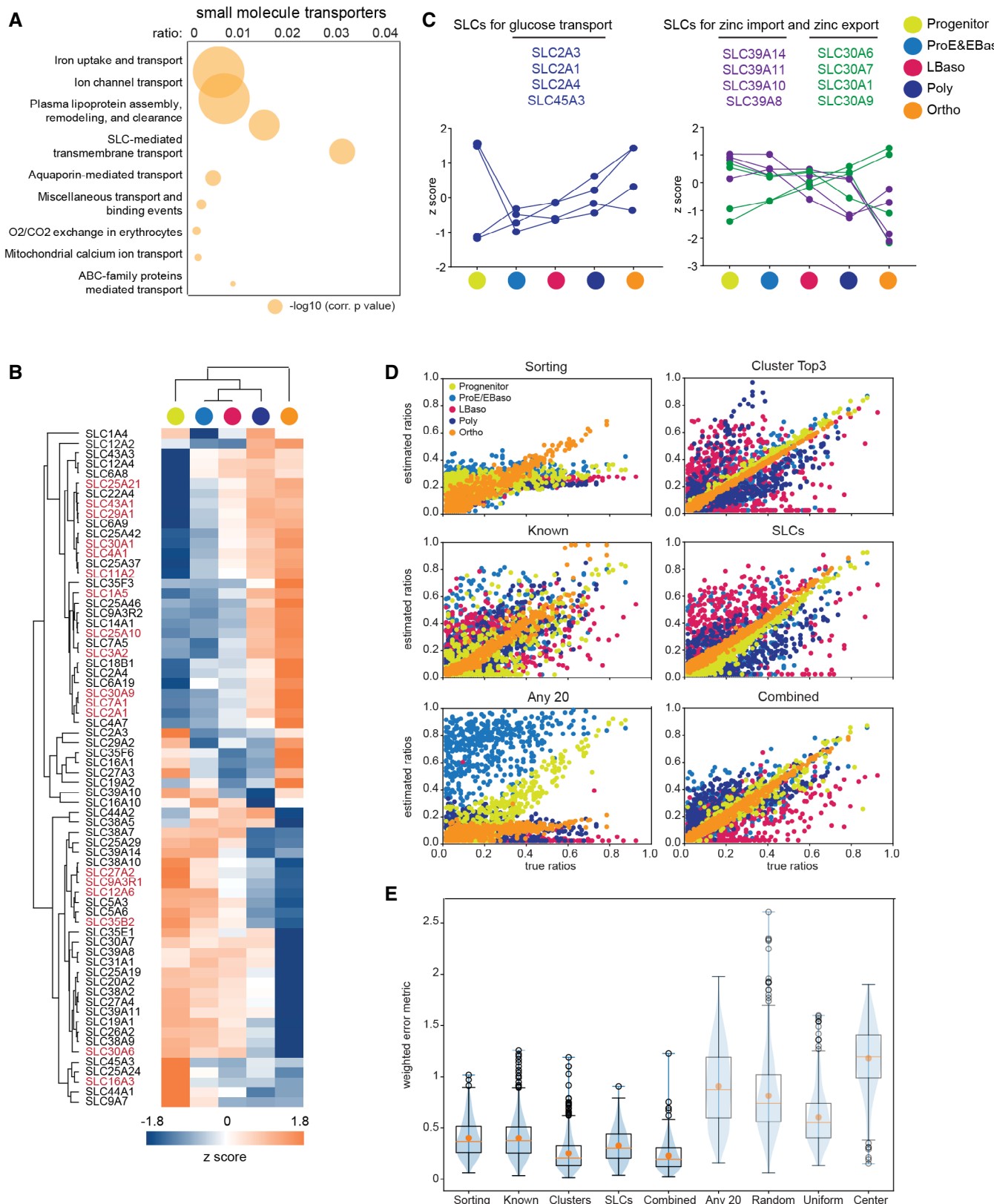

**Figure 3.**

◀

**Figure 3.  Solute carriers in erythroid maturation and computational extraction of stage-specific protein markers.**

A   Plot shows the overrepresented Reactome pathways (Jassal, 2011; Jassal, Matthews *et al*, 2020) with their corrected *P*-values (-log10) and the ratios of given entities from a particular pathway versus all entities from that pathway ($n = 1,882$).

B   Heat map of z-scored SLC protein abundances (log2 DIA intensities) across differentiation. The proteins in red were used to generate the input matrix for the SLCs marker set used in (D and E).

C   Expression of four different glucose transporters during erythrocyte development (log2 DIA intensities, left panel). Countervailing expression regulation of zinc importers and exporters during erythrocyte development (log2 DIA intensities, right panel).

D   Computation sorting quality comparing predefined versus estimated ratios of cells in the five differentiation stages.

E   Accuracy of cell type prediction based on different protein marker sets (x-axis) as measured by a weighted error metric (y-axis) across the 500 generated sets of mixture ratios (also see Materials and Methods). The blue violin plots illustrate the underlying distribution reaching from minimum to maximum. The black box plots depict the quartiles of the distribution with whiskers extending to the quartiles $\pm$ 1.5 $\times$ interquartile range. The orange horizontal lines indicate the median, and the orange dot highlights the mean of the distribution.

(Table EV1). In addition, our dataset also contains many transporters—including for vitamins, lipids, and whose substrates have not yet been identified—vastly extending the repertoire of SLCs and transported molecules associated with erythropoiesis.

Among the prominent observations emerging from our data were the several differentially expressed SLCs attributed to a common ligand. We first focused on hexose/glucose transporters. Erythrocytes obtain energy only through glycolysis, which makes glucose uptake crucial for erythrocyte development and mature erythrocytes (Guizouarn & Allegrini, 2020). Glycolysis provides ATP, essential to fuel ion pumps, and NADH, a powerful reducing agent that prevents heme oxidation. SLC2A1/GLUT1 is thought to be the main transporter for glucose in erythrocytes, but also imports L-dehydroascorbic acid (DHA), which is reduced to ascorbic acid and essential for maintaining reductive capacity in plasma (Montel-Hagen, Kinet *et al*, 2008). Two out of four identified transporting glucose SLCs, SLC2A1/GLUT1 and SLC2A4/GLUT4, gradually increased from progenitors to Ortho (Fig 3B and C, Table EV1), supporting the idea of increased uptake of DHA by SLC2A1/GLUT1 during erythroid maturation and glucose import in mature erythrocytes. Interestingly, SLC2A4/GLUT4, primarily known to be highly expressed in skeletal muscle and playing a key role in regulating blood glucose concentration (Huang & Czech, 2007), might be important in erythrocytes as well, contributing to glucose uptake in addition to SLC2A1/GLUT1. The other two glucose transporters (SLC2A3 and SLC45A3) are highly expressed specifically in progenitors and to our knowledge have not been associated with erythropoiesis; their regulation would be interesting to investigate in the future.

A second remarkable example concerns SLCs for transporting metal ions (14 identified in total) with a full eight of them dedicated to zinc import and export. Maintenance of intracellular zinc levels controlled by GATA/heme circuit has recently been discovered as a vital determinant of erythroid maturation (Tanimura, Liao *et al*, 2018). This indicates the adaptation of differentiating cells to stage-specific metabolic requirements and their interaction with the environment. Apart from the zinc importer SLC30A1 and zinc exporter SLC39A8, previously described in a "zinc switch" model reflecting their reciprocal expression during terminal erythropoiesis (Tanimura *et al*, 2018), we here uncovered additional three upregulated exporters (SLC39 family) and three downregulated importers (SLC30 family) during maturation (Fig 3B and C, Table EV1). Only one of these had prior implications in erythrocyte homeostasis (Ryu, Lichten *et al*, 2008), suggesting even more intricate and possibly redundant regulation of zinc homeostasis.

## Computational extraction and characterization of stage-specific protein markers

Given the distinct stage-specific expression patterns of the SLCs, we wondered if they could even serve as marker and selection proteins. The standard approaches for distinguishing erythroid developmental stages rely on canonical cell surface markers, including the ones we employed for FACS enrichments (Chen *et al*, 2009). Our proteomics analysis revealed that drastic proteome-wide changes of numerous proteins occurred at transitions, in particular from progenitors to ProE/EBaso and from Poly to Ortho, implying the expression of numerous new stage-specific protein markers that might be exploited for refining the isolation and quantification of each differentiation stage. Panels of proteins with characteristic profiles could also be useful for *in silico* deconvolution of mixed developmental populations.

As a starting point, we investigated four protein sets for their ability to distinguish the different developmental stages. The first two are a *known marker* set of 22 proteins as well as the *FACS sorting markers* (Table EV1). These proteins correlated well with their expected expression profiles along the differentiation process. We next constituted a third marker set of 18 *SLCs* on the basis of the most consistent quantification profiles (Fig 3B). As a final set, we selected 18 stage-specific proteins from our proteomics data comprising the top three most significant ones for each of the six clusters in Fig 2A (*cluster Top3* set, smallest ANOVA *q*-values) (Table EV1). Among those, KLF13, which activates the promoters of several erythroid genes *in vitro*, was gradually upregulated until very late stages, consistent with its reported role in mouse erythroblast maturation (Asano, Li *et al*, 2000; Gordon, Outram *et al*, 2008).

With these four protein panels in hand (*sorting and known markers, cluster Top3, SLCs*), we set out to compare their ability to distinguish different developmental stages. Briefly, we generated 500 random *in silico* mixtures of aggregated protein abundances from a linear combination of the five differentiation stages. The mixtures were prepared at predefined ratios. Each of the four marker panels was subsequently used to estimate the true mixing ratios. The agreement between the estimated and true, predefined ratios of the *in silico* mixtures was used to evaluate how well each of the four marker panels can distinguish the different developmental stages (Fig 3D and E).

The *sorting and known markers* reasonably estimated the fraction of the Ortho stage in the mixtures, but performed worse for all other stages (diagonal orange markers in Fig 3D). Remarkably,

the *cluster Top3 and SLC markers* better characterized the differentiation process than previously known proteins and produced more accurate estimations for both Ortho and progenitor fractions in the computational mixture populations (diagonal orange and yellow markers in Fig 3D). However, they were still less effective at distinguishing stages from ProE to Poly, in line with their smaller proteome differences in our data (Fig 3D). A combined set of 62 proteins outperformed all others, even in estimating intermediate, adjacent differentiation stages as judged by a quantitative error analysis and compared to random controls (Methods, Fig 3E). In addition to recent advances in single cell transcriptomics (Tusi *et al*, 2018), our deconvolution approach could further aid the identification of specific populations among bulk pools obtained during erythropoiesis, for example, in the study of differentiation dynamics from *in vivo* samples. The proteins selected in this analysis, especially the *SLCs,* add to our resource as they are interesting candidates for investigating stage-specific mechanisms in follow-up studies.

## An orchestrated network of erythropoietic kinases and their downstream target

Several kinases act or have already been implicated in a complex regulatory network in erythropoiesis. To advance our understanding of the dynamic phospho-regulatory network during erythropoiesis, we assessed temporal kinase activities at a global scale across terminal maturation. Mining of our proteome data revealed an astounding 270 kinases and 90 phosphatases that were differentially expressed with clear stage-specific profiles during differentiation (Fig 4A). To investigate their activities, we turned to phosphoproteomics which globally captures their substrates (Fig 4B). We enriched phosphopeptides from the same differentiation stages in biological quadruplicates using the EasyPhos platform (Fig 4B) (Humphrey *et al*, 2015; Humphrey *et al*, 2018). This streamlined protocol enabled deep profiling of phosphoproteomes at specific developmental stages in single-run DDA measurements from only 80 µg of protein lysates, capturing 27,166 distinct phosphosites on more than 4,200 proteins (Fig 4C). Almost 20,000 sites were identified in more than two replicates of at least one maturation stage and 3,604 were novel sites according to the PhosphoSitePlus database (Hornbeck, Kornhauser *et al*, 2012) (Fig 4C and Dataset EV2). Given the prominent changes in the plasma membrane proteome, it was interesting to see that 401 of them had phosphosites. This encompassed 23 of the aforementioned SLCs, suggesting stage-specific signaling roles (Taylor, 2009) in addition to their dynamic

expression across stages. Specifically, our phosphoproteomics also identified Ser/Thr phosphorylation sites on Band 3, whose tyrosine phosphorylation is known to enable docking of cytoplasmic signaling molecules (Yannoukakos, Vasseur *et al*, 1991; Brunati, Bordin *et al*, 2000).

For further statistical analysis, we used a stringently filtered dataset of 12,216 phosphopeptides quantified in all four replicates of at least one differentiation stage. Strikingly, about half of these phosphosites significantly changed in at least one developmental transition (ANOVA, FDR < 0.05) and a quarter of all phosphosites (3,089) were dephosphorylated from Poly to Ortho stage (Fig 4D and Dataset EV2).

To compare the dynamics of the phosphoproteomes to the proteomes, we visualized fold change distributions of quantified proteins (gray) and phosphopeptides (pink) for three pairwise comparisons: (i) progenitor versus ProE/EBaso, (ii) ProE/EBaso versus Poly, and (iii) Poly versus Ortho (Fig 4E). The fold change distributions of phosphopeptides were considerably more scattered than those of proteins in all three comparisons, reflecting dynamic, large-scale phosphoregulation. The largest fold change of regulated phosphopeptides occurred between early stages of progenitor to ProE/EBaso and the later stages, Poly to Ortho (Fig 4E). The highly dynamic changes in global phosphorylation landscape likely reflect critical roles for distinct kinases at specific maturation stages.

Next, we inferred kinase activities from the phosphoproteome by stage-dependent enrichment analysis using PhosFate profiler (Fig 4 F) (Ochoa, Jonikas *et al*, 2016). This method predicts changes in kinase activity by testing the enrichment of differentially regulated, annotated kinase-substrate motifs. Substrates peaking during the early stages of differentiation (ProE/EBaso) were enriched with motifs for kinases of the MAPK signaling network (BRAF, MAPK1, MAPK3, FYN, SRC), which are known to promote cell cycle and proliferation (Carroll *et al*, 1991; Sakamoto, Kitamura *et al*, 2000; Geest & Coffer, 2009). Interestingly, the observed substrate phosphorylations suggest that CDK1 and many other cell cycle associated kinases (AURKB, BUB1, CDK14, CDK16, CDK2, CDK3, CDK4, CDK5, CDK6, and DYRK3) remain active until very late stages (Poly and Ortho). Unexpectedly, DNA damage checkpoint kinases (ATM, ATR, and CHEK2) were also enriched. During terminal maturation, erythroblasts undergo a series of specialized cell divisions associated with shortened G1 phase and increased lineage-specific gene expression (Hwang, Hidalgo *et al*, 2020). It is possible that DNA damage checkpoint activity is triggered to control replicative stress-induced damage during this process, similar to what occurs upon

---

**Figure 4.  An orchestrated network of erythropoietic kinases and their downstream targets.**

A   Heat map of z-scored and differentially regulated kinase and phosphatase abundances (log2 DIA intensities) across differentiation.

B   Experimental design of the phosphoproteomic study, performed on the same populations as collected for the full proteome analyses (also see Figure 1A). Analytical workflow including phospho-enrichment, single shot DDA acquisition, and data analysis.

C   Number of identified and quantified Class 1 phosphosites (localization probability to a single amino acid > 0.75) after filtering for 50% data completeness in at least one differentiation stage. Total number of phosphoproteins is also shown.

D   Significantly regulated phosphorylated sites in pairwise comparisons of ProE/EBaso versus Progenitor, Poly versus ProE/EBaso, and Ortho versus Poly.

E   Distributions of phosphopeptides and their matching proteins based on their log10 intensities (y-axis) versus log2 test differences (x-axis) are illustrated for ProE/EBaso versus Progenitor (left), Poly versus ProE/EBaso (middle), and Ortho versus Poly (right). Pink represents phosphopeptides, whereas gray represents proteins.

F   Stage-specific predicted active kinases based on targeted sites identified by PhosFate profiler (http://phosfate.com). Left boxes represent kinases whose substrates are preferentially detected at the earlier stage of differentiation and right boxes represent those whose substrates are preferentially detected at the later stage.

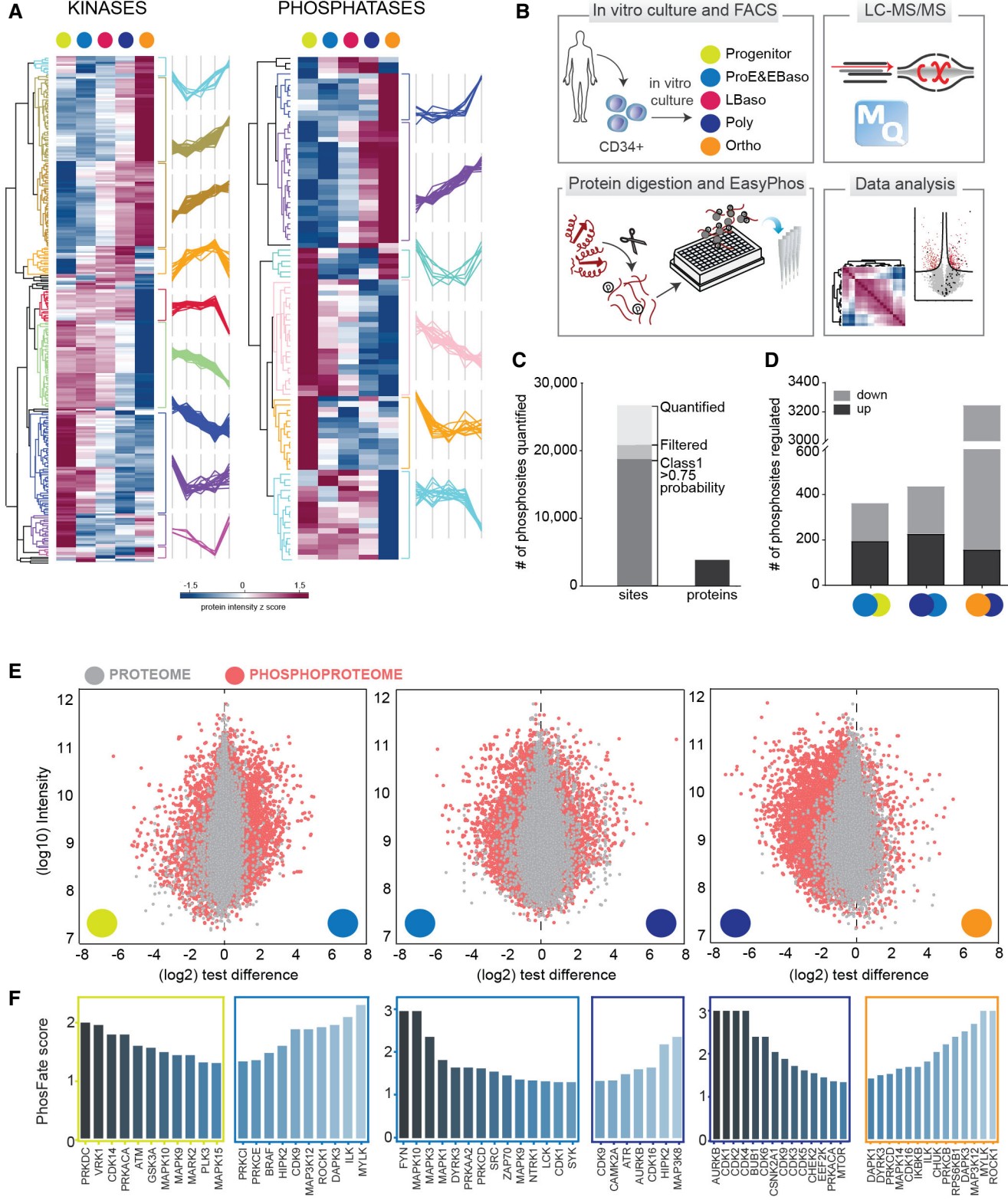

Figure 4.

reactivation of dormant HSCs (Walter, Lier *et al*, 2015). Erythroid differentiation is also accompanied with increased metabolic activity and the production of reactive oxidative species (ROS) which might lead to DNA damage with checkpoint kinase activation. Together, our data reveal a rich network of temporally activated kinases during differentiation of human erythrocytes.

## CRISPR/Cas9 screen reveals critical functions of the erythropoietic kinome

The proteomics analysis established a "kinome atlas" revealing dynamic changes in kinase abundance and activity at distinct stages of erythropoiesis, with a dramatic decrease in the global phosphoproteome during late maturation. To investigate potential functional implications of these changes, we performed a CRISPR/Cas9 screen in HUDEP-2 cells, an immortalized human erythroblast line that proliferates in an immature state and can be induced to undergo terminal maturation by manipulation of culture conditions (Kurita, Suda et al, 2013). HUDEP-2 cells stably expressing Cas9 (HUDEP-2$^{Cas9}$) were transduced at low multiplicity of infection with a lentiviral vector library encoding 3,051 single guide (sg) RNAs targeting the coding regions of most known kinases ($n = 482$) and a green fluorescence protein (GFP) reporter gene (Grevet, Lan et al, 2018; Tarumoto, Lu et al, 2018b) (Figs 5A and EV5A–D). Two days later, GFP$^+$ cells were flow cytometry-purified, and split into pools for further expansion or induced maturation, followed by next-generation sequencing (NGS) to assess sgRNA abundance (Fig 5A). Compared to cells at 2 days post-transduction ("day0"), 30 sgRNAs were underrepresented after 10 days of expansion (FDR < 0.05), reflecting candidate kinase genes that promote survival and/or proliferation of immature erythroblasts (Fig 5B and Table EV2). These genes encoded cyclin-dependent kinases (CDK1, CDK7, CDK9), cell signaling components (KIT, JAK2), DNA damage checkpoint response proteins (ATR, CHEK1), and a regulator of ion flux (OXSR1). Several of these genes exhibited maturation stage-specific expression in our proteome analysis (e.g., CDK1, CDK9, ATR, KIT, and JAK2) (Fig 4F). The KIT and JAK2 genes are essential signaling molecules for erythropoiesis (Neubauer, Cumano et al, 1998; Parganas, Wang et al, 1998; Munugalavadla & Kapur, 2005a). Previous proteomic studies identified OXSR1 (OSR, oxidative stress-responsive kinase 1) as one of the most abundant Ser/Thr kinases in reticulocytes and mature erythrocytes (Gautier, Leduc et al, 2018). The OXSR1 protein phosphorylates Na$^+$–K$^+$ and K$^+$–Cl$^-$ membrane co-transporters to activate and inhibit their activities, respectively (de Los Heros, Alessi et al, 2014). Our data suggest a role for OSXR1 in the maintenance of erythroid precursors.

Transduced HUDEP-2 cells induced to undergo terminal maturation were cultured for 3 days, fractionated according to their expression of the late-stage erythroid marker Band 3, and analyzed by NGS for sgRNA abundance. Single guide RNAs for five genes were significantly overrepresented in immature (Band3$^-$) cells, indicating that these genes are positive effectors of maturation, while sgRNAs for nine genes were overrepresented in mature (Band3$^+$) cells, representing candidates that inhibit maturation (Fig 5C and Table EV3). There was minimal overlap between genes that affect expansion or maturation (Fig EV5E). Notably, eight kinases identified as regulating differentiation in the CRISPR/Cas9 screen are also identified among the stage-specific active kinases predicted by phosphorylation of their cognate motifs (Figs 4F and EV5F).

We noted that disruption of numerous genes stimulating the Ras/MAPK signaling pathway caused accelerated erythroid maturation (Fig 5C). Three of the corresponding proteins, RAF1, BRAF1, and MAPK1, are members of the canonical Ras/MAPK family, while LYN is known to engage and potentiate RAF1 (Tilbrook, Colley et al, 2001). In non-erythroid cells, PIM1 kinase has been shown to phosphorylate ERK and activate Ras/MAPK signaling (Wang, Anderson et al, 2012). To validate these candidates, we transduced Cas9-expressing HUDEP-2 cells with individual sgRNAs for each gene, followed by cell culture in differentiation medium or expansion medium. We achieved at least 50% insertion–deletion (indel) mutation frequency for each sgRNA (Fig EV6A). Consistent with results of the CRISPR screen, disruption of RAF1, BRAF1, MAPK3 (ERK2), LYN, or PIM1 resulted in significantly accelerated terminal maturation (Figs 5D and E, and EV6B–D). Additionally, disruption of MAPK3, LYN, and PIM1 appeared to inhibit cell expansion slightly (Fig EV6B and D). Of note, disruption of MAPK3 and LYN was not observed to inhibit cell growth in the CRISPR screen. Thus, these studies will require further validation, including assessment of additional sgRNAs and their potential off-target effects. However, our findings are consistent with the results of the CRISPR screen in general, and indicate that downregulation of individual Ras/MAPK/ERK pathway components promotes terminal erythroid maturation.

**Figure 5. Kinome-targeting CRISPR/Cas9 screen in HUDEP-2 cells.**

A    Workflow of a CRISPR/Cas9 screen with an sgRNA library targeting 482 human kinase genes to identify those that alter erythroid precursor expansion or terminal maturation.

B    Volcano plots showing FDR versus log2 fold change in sgRNA abundance between day 0 and day 10 of expansion. Results were analyzed using MAGeCK (Materials and Methods). Each dot represents a single kinase gene based on the enrichment of four sgRNAs. Significantly different genes (FDR < 0.05; log$_2$ fold change<−1) are shown in red.

C    Volcano plots showing FDR versus log2 fold change in sgRNA abundance between Band3 high and low fractions after 3 days of maturation. Results were analyzed using MAGeCK (Materials and Methods). Each dot represents a single kinase gene based on the enrichment of four sgRNAs. The significant positive or negative regulators for maturation (FDR < 0.05; log2 fold change<−0.5 and log2 fold change> 0.5) are shown in purple and green, respectively.

D    HUDEP-2 cells expressing Cas9 were transduced with lentiviral vectors encoding single sgRNAs targeting the indicated genes, induced to undergo terminal maturation, and analyzed after 3 days. Graph shows fraction of Band3$^+$ cells. Error bars represent mean ± SEM of 3 biological replicates. *P < 0.05, **P < 0.01, ***P < 0.005; n.s., not significant; unpaired t-test.

E    Effects of two different PIM1-targeting sgRNAs on erythroid maturation of HUDEP-2 cells, performed as described for panel (D). Error bars represent mean ± SEM of 3 technical replicates. ****P < 0.0001; unpaired t-test.

F    Protein abundances (log2) of PIM1 during terminal differentiation of primary erythroblasts. Yellow highlighting indicates that SCF and EPO were present in the culture medium, while pink indicates EPO only. The mean values (±SEM) of normalized intensities to the median of four replicates of the progenitor stage are shown.

G    Heat map showing z-scored (log2) phosphopeptide intensities detected in known PIM1 substrates.

H    Consensus PIM1 kinase motif from PhosphoSitePlus database (Hornbeck et al., 2012).

I    Heat map of z-scored (log2) phosphopeptide intensities of potential PIM1 kinase targets identified by motif analysis.

J    Gene Ontology (GO) enrichment analysis of potential PIM1 kinase targets, performed using Fisher's exact test. 5% threshold was applied to Benjamini–Hochberg FDR to determine the significance.

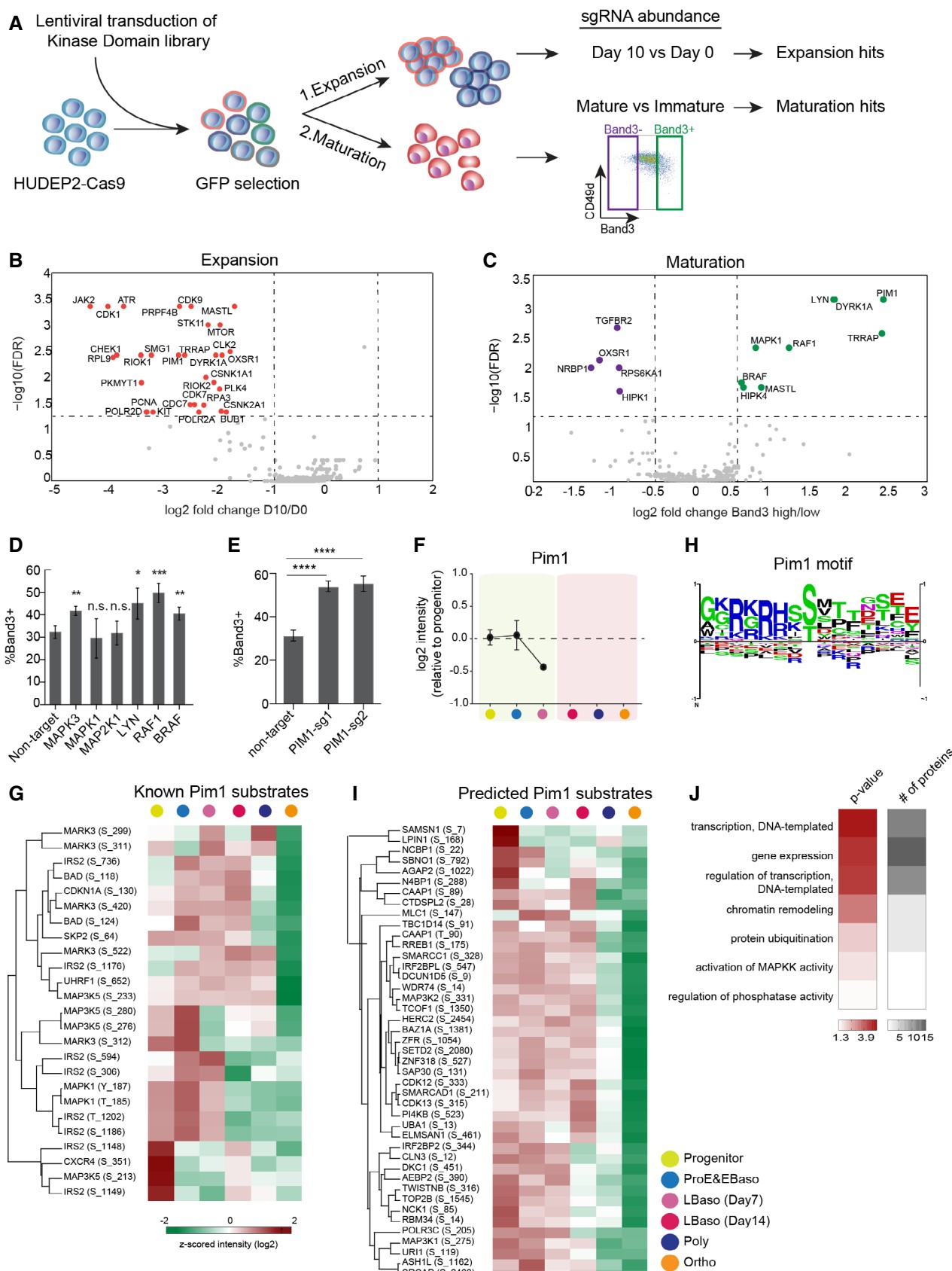

**Figure 5.**

In agreement with this model, the *TGFRB2* gene, identified as positive regulator of maturation (Fig 5C), is known to inhibit MAPK signaling (Li, Li *et al*, 2014).

Consistent with the result of the screen that PIM1 is a candidate that inhibits erythroid maturation (Fig 5C), PIM1 protein levels were undetectable after the LBaso stage (Fig 5F). The *PIM1* gene encodes a serine–threonine kinase oncoprotein that stimulates cell survival and cell cycle progression by phosphorylating numerous substrates that have been identified in non-erythroid cell types. To identify potential effectors of PIM1 signaling in erythroid cells, we searched our phosphoproteomics data for previously described PIM1 substrates (Hornbeck *et al*, 2012) and identified 25 significant phosphosites (out of 100) that coincided with PIM1 expression (Fig 5G). We then generated a PIM1 phosphorylation site consensus motif using the PhosphoSitePlus database (Hornbeck *et al*, 2012) (Fig 5H), and investigated whether the motif is enriched in the erythroid maturation stage-dependent phosphoproteome. This identified 79 phosphorylation targets of which 54% have maturation stage-dependent profiles correlating with PIM1 protein abundance (Fig 5 I). GO-term analysis revealed significant enrichment of terms associated with "chromatin remodeling", "transcriptional regulation", "kinase/phosphatase activity", and "ubiquitylation" (Fig 5J). Of particular interest were Ras/MAPK family members such as MAP3K1, MAP3K5, and MAP3K2, consistent with a regulatory role of PIM1 in Ras/MAPK signaling.

## System-wide dissection of c-Kit and EPOR phosphosignaling in erythropoiesis

Phosphoprotein analysis and a CRISPR/Cas9 screen defined a dynamic, developmental stage-specific kinome during erythropoiesis and indicated that Ras/MAPK downregulation is critical for erythroid maturation. We explored this further by examining our proteomics dataset for Ras/MAPK signaling components in relation to the expression and activities of c-Kit and EPOR. The kinetics of Ras/MAPK protein expression varied across erythroid maturation with MAPK1 (ERK2), MAPK3 (ERK1), and RAF1 persisting into late maturation stages (Fig 6A), suggesting that their activity may be suppressed post-translationally. In agreement, activating T185/Y187 phosphorylations on MAPK1 (ERK) indicated maximal activity during ProE/EBaso and termination by the LBaso stage (Fig 6B) (Michaud, Fabian *et al*, 1995; Gupta & Prywes, 2002). The activating S63 phosphorylation on ATF1, a distal target of Ras/MAPK signaling, peaked later (at the LBaso stage) and persisted throughout erythroid maturation (Fig 6B). The RSK kinase, which phosphorylates ATF1, is activated by both MAPK and PI3K/AKT/mTOR signaling (Koh, Jee *et al*, 1999).

The erythroid cytokine receptors c-Kit and EPOR have distinct roles in erythropoiesis, although their signaling pathways overlap considerably (Fig 6C). Our erythroid culture system contained both SCF (c-Kit ligand) and EPO in the first and second phase (day 0–12) and EPO only in the third phase (days 12–14) (Fig 1A). The rationale for this culture system is based on findings that persistently elevated SCF-c-Kit signaling inhibits terminal erythroid maturation (Muta, Krantz *et al*, 1995; Munugalavadla, Dore *et al*, 2005; Haas, Riedt *et al*, 2015). The levels of c-Kit and EPOR/JAK2 proteins decreased during differentiation but with differing kinetics (Fig 6D). c-Kit levels declined after the ProE stage, similar to ERK activity. Tyrosine phosphorylation in the c-Kit cytoplasmic domain, which reflects receptor activity (Lennartsson & Ronnstrand, 2012), was maximal in erythroid progenitors and ProE/EBaso and decreased by the LBaso stage, even with SCF present in the culture media (Fig 6E and F). ERK2 phosphorylation declined at the same stage, even with SCF present in the culture medium (LBaso day 7) and was undetectable in LBaso day 14 when SCF was not present in the medium. Thus, c-Kit protein levels and its phosphorylation, along with downstream Ras/MAPK signaling, are downregulated relatively early in erythroid maturation, consistent with the literature (Matsuzaki, Aisaki *et al*, 2000; Gautier *et al*, 2016).

Compared to c-Kit, the expression of EPOR/JAK2 declined more slowly during erythropoiesis and was sustained to later stages. We were not able to detect phosphorylation of EPOR or JAK2, perhaps because the levels of these proteins are relatively low. Compared to ERK2, STAT5 phosphorylation declined more slowly and persisted until the later stages of erythropoiesis, similar to the kinetics of EPOR expression (Fig 6G). Thus, ERK2 phosphorylation levels parallel the expression and activation of c-Kit, while phospho-STAT5 levels correlate with expression of EPOR, likely reflecting preferential signaling activities of the two cytokine receptor pathways. Together, our findings suggest that Ras/MAPK activity occurs

**Figure 6. System-wide dissection of c-Kit and EPOR phosphosignaling.**

A   Protein abundances (log2 DIA intensities) normalized to progenitor stage. In all panels, stages shaded yellow were cultured with SCF and EPO, while those shaded pink were cultured with EPO only. Data is plotted if quantified in at least 50% of biological replicates. Error bars represent mean ± SEM of at least two biological replicates.

B   Profiles of phosphorylations (log2 DDA intensities) normalized to progenitor stage. The mean values (±SEM) of normalized phosphopeptide intensities to the median of four replicates of the progenitor stage are shown.

C   Major signaling pathways downstream of c-Kit and EPOR activation by their corresponding ligands stem cell factor (SCF) and erythropoietin (EPO). Shaded genes indicate those indentified to inhibit erythroid maturation in the CRISPR/Cas9 screen described in Figure 5.

D   Protein abundances (log2 DIA intensities) of c-Kit, EPOR, and JAK2, normalized to progenitor stage. The mean values (±SEM) of normalized protein intensities to the median of four replicates of the progenitor stage are shown.

E   Following activation by SCF, phosphorylated tyrosine residues on c-Kit receptor serve as binding sites to key signal transduction molecules (SRC, GRB2, and PI3K) resulting in activation of downstream signaling pathways.

F   Heat map of z-scored (log2) phosphopeptide intensities of c-Kit receptor.

G   Profiles of STAT5A/B phosphorylations (log2 DDA intensities) normalized to progenitor stage. The mean values (±SEM) of normalized phosphopeptide intensities to the median of four replicates of the progenitor stage are shown.

H   Heat map of z-scored protein abundances (log2 DIA intensities) of phosphatases. In (A, B, D, and G) panels, dashed line represents a reference line to determine the fold change compared to the progenitor stage.

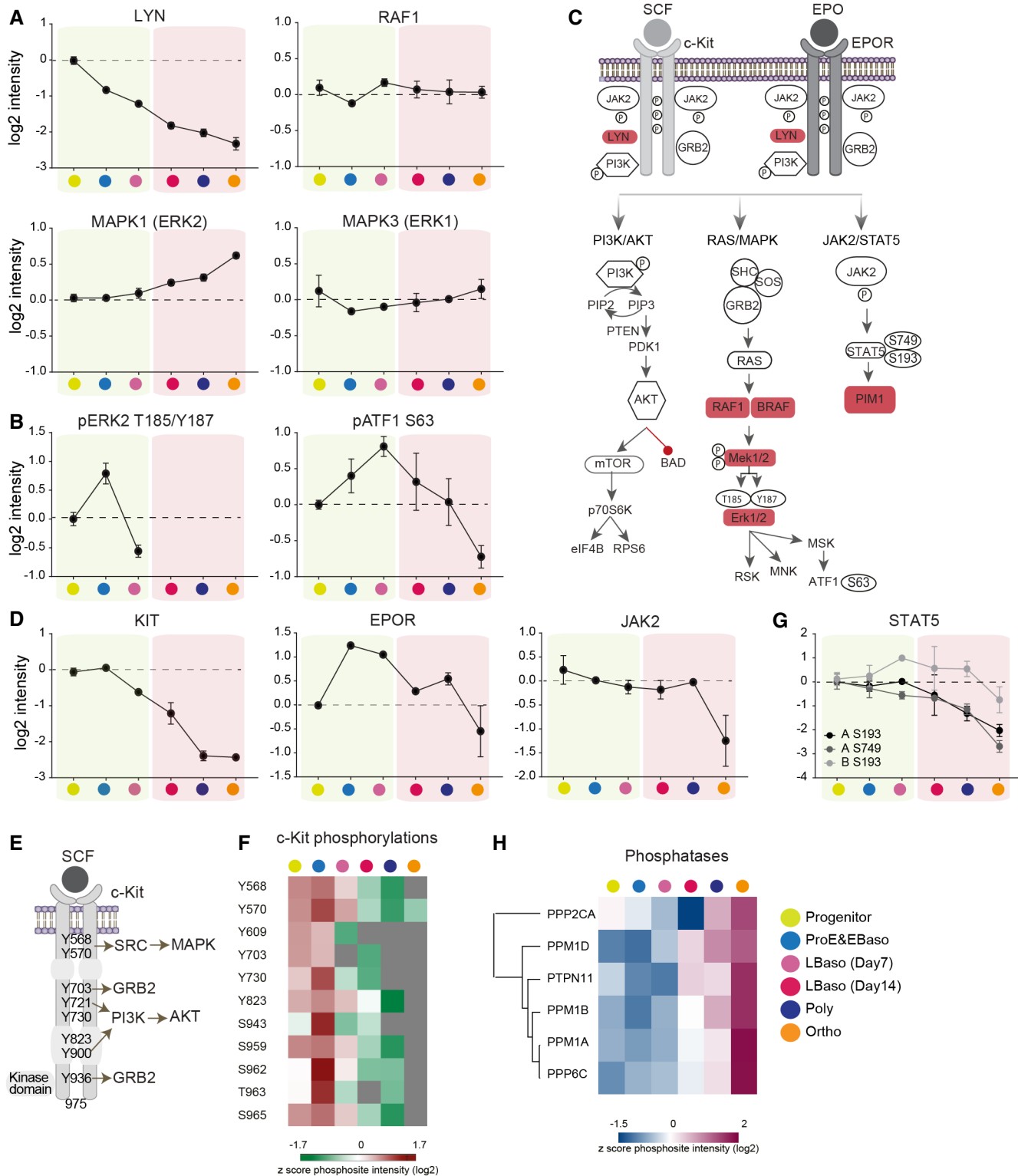

**Figure 6.**

during early stages of erythropoiesis, delays terminal maturation and is c-Kit driven. This is consistent with the established role for c-Kit in supporting proliferation and survival of early erythroid progenitors and a requirement for c-Kit downregulation during

normal erythropoiesis (Nocka *et al*, 1989; Bernstein, Forrester *et al*, 1991; Muta *et al*, 1995).

Phosphorylation is also regulated by phosphatases. Of 51 phosphatases implicated in inhibiting the Ras/MAPK pathway (Kondoh

& Nishida, 2007; Li, Wilmanns *et al*, 2013), 16 were detected in our data, 6 of which were induced during terminal maturation (Fig 6H). The majority of these phosphatases are novel candidate genes whose roles in erythropoiesis need further exploration.

## Discussion

Here we show that in-depth quantitative proteomic and phospho-proteomic analyses of purified erythroid precursors at distinct maturation stages is now made possible by state-of-the-art MS-based proteomics and our EasyPhos technology, allowing us to assess erythroid maturation at the level of the proteins, the main functional cellular entities. The breadth and depth of coverage achieved by these technologies offer unbiased system-wide insights into the regulation of erythropoiesis, which we complemented further by performing an unbiased CRISPR/Cas9 screen to interrogate the erythroid kinome.

Our findings derived from analysis of unbiased global proteome and phosphoproteome data can be linked to several aspects of erythroid biology. Dramatic alterations were observed at the transitions from progenitor to Pro/EBaso, coincident with sharp reductions in proliferation capacity, transition of signaling dependency from SCF/c-Kit to EPO/EPOR and the onset of an erythroid-specific gene expression program mediated by transcription factors GATA1 and TAL1. Another striking change in the proteome and phospho-proteome was detected at the transition of Poly to Ortho, which immediately precedes a dramatic streamlining of the proteome, including elimination of most organelles and degradation of unnecessary cytosolic proteins. Further proteomic studies of erythro-poiesis focusing on post-translational modifications such as ubiquitylation may provide further insights into the development of the highly specialized red blood cell proteome. For example, the E2/E3 ubiquitin ligase UBE2O mediates the proteasomal clearance of numerous erythroid proteins and whole ribosomes during reticulo-cyte maturation (Nguyen *et al*, 2017; Yanagitani, Juszkiewicz *et al*, 2017). We also observed large-scale downregulation of phosphorylation events during late-stage erythroid maturation. The underlying mechanism is unknown, but likely based on the overall deactivation and elimination of most kinases and/or the upregulation of phosphatase activity(ies). This process may also conserve ATP for phosphorylation of a few erythroid-specific target proteins, such as Band 3, Band 4.1, and GYPA (Boivin, 1988; Harrison, Rathinavelu *et al*, 1991; Manno, Takakuwa *et al*, 2005).

The analyses of proteins mediating solute transport and phospho-based signaling highlight two examples by which the data can be mined for hypothesis generating discovery and focused problems related to erythroid biology. The key concept is that proteome-wide changes accompanying differentiation from early erythroid progenitors into nearly mature erythrocytes involve remarkable regulation of and by signaling pathways—at the protein level.

Tracking the levels across particular families of proteins defined numerous distinctive stage-specific profiles, exemplified by coordinated expression of specific cohorts of SLCs, kinases, and phosphatases. For some of these SLCs, previous reports established roles in transporting crucial molecules for erythropoiesis (Dataset EV2). However, the much larger repertoire of SLCs of unknown biological function likely reflects erythroid stage-specific metabolic

requirements to be elucidated in future studies. Beyond this, both the substrate diversity of SLCs and their widespread phosphorylation (average of three per SLC) point toward exquisite fine-tuning and coordination of transporters with stage-specific signaling pathways. So far, very few examples of this mode of regulation have been described, most prominently the tyrosine phosphorylation of the SLC Band 3, which mediates docking of cytoplasmic signaling molecules (Brunati *et al*, 2000). The prevalence of stage-specific SLC phosphorylation observed here may indicate system-wide coordination of small molecule transport with cytoplasmic signaling throughout erythroid maturation and across many transporters. SLCs are relevant to therapeutics of several human diseases and to drug discovery, either as drug targets themselves or as mediators of drug uptake (Cesar-Razquin *et al*, 2015). It now seems likely that the varying expression of SLCs with overlapping selectivities, as well as their regulation by post-translational modification, will also contribute to pathology and provide opportunities for therapeutic development (Noomuna, Risinger *et al*, 2020). Importantly, our data provide a framework for system-wide studies of SLC small molecule flux and signaling throughout the differentiation process. The dynamics of SLCs, together with more than 700 other quantified membrane proteins may furthermore contribute to our understanding of changing cell membrane properties required for erythropoiesis.

The distinct cohorts of kinases and phosphatases expressed coordinately and with varying kinetics across the erythrocyte maturation pathway likewise reflect extensive protein-level regulation, in this case through post-translational modification. With this notion in mind, we complemented the quantitative stage-specific proteome measurements with a kinome-targeting CRISPR/Cas9 screen and phosphoproteomics, which provides a profile of system-wide signaling across erythroid maturation. PIM1, which has previously been shown to be a downstream target of STAT5 following EPOR and c-Kit activation (Menon, Karur *et al*, 2006; Gillinder, Tuckey *et al*, 2017), was identified as the highest scoring hit for maturation of HUDEP-2 cells. Pursuing PIM1, we defined a composite profile based on (i) stage-specific expression, (ii) phosphorylation kinetics of known substrates, and (iii) a PIM1 consensus sequence by correlation. Further analysis then provided a list of candidate PIM1 substrates that kinetically parallel PIM1 activity and may coordinate PIM1 activity with that of other diverse signaling effectors including epigenetic regulators, regulators of ubiquitin signaling, and more, whose functional roles in erythropoiesis can now be studied. The proposed function of PIM1 as a ribosomal stress sensor further suggests a possible role for these substrates in Diamond–Blackfan anemia, an inherited ribosomopathy that causes pure red cell aplasia (Iadevaia, Caldarola *et al*, 2010; Sagar, Caldarola *et al*, 2016).

We took advantage of our data to mine the stage-resolved phosphoproteomics of the SCF- and EPO-triggered signaling networks, which are critical to erythropoiesis. Combined results from phosphoproteomics and a functional CRISPR/Cas9 screen highlight a general decrease in kinase activity across the erythroid proteome during terminal maturation, with a sequential downregulation of c-Kit and EPOR activities. In general, c-Kit signaling is thought to drive erythroid precursor proliferation, while EPOR signaling drives terminal maturation (Munugalavadla & Kapur, 2005b). Although both receptors activate MAPK/ERK1/2, the effects of c-Kit appear to

be more potent (Pircher, Geiger *et al*, 2001) and numerous studies indicate that downregulation of this activity is essential for normal erythropoiesis. For example, erythroid maturation is inhibited by sustained MAPK and ERK1/2 activation caused by an activating c-Kit mutation (Haas *et al*, 2015), oncogenic RAS proteins (Zhang & Lodish, 2004; Zhang, Liu *et al*, 2007), TRAIL receptor activation (Secchiero, Melloni *et al*, 2004), *BCR-ABL* fusion protein (Kawano, Horiguchi-Yamada *et al*, 2004), and PIEZO ion channel activation (Caulier, Jankovsky *et al*, 2020). Conversely, inactivation of ERK accelerates erythroid maturation in K562 cells (Kawano *et al*, 2004) and *Erk1*$^{-/-}$ mice exhibit amplified splenic erythropoiesis and enhanced maturation of erythroid precursors (Guihard, Clay *et al*, 2010). Our findings support this model by showing that c-Kit activity and ERK2 phosphorylation are downregulated prior to the loss of EPOR/JAK2 activity and that erythroid maturation is accelerated by inhibition of MAPK/ERK signaling components in an unbiased CRISPR/Cas9 screen. In addition, the CRISPR/Cas9 screen demonstrated potential roles for *PIM1* and *TGFRB2* in erythroid maturation, perhaps via modulation of MAPK/ERK signaling.

Functional interactions between c-Kit and EPOR, and the regulation of their expression and signaling during erythropoiesis are not fully understood. The c-Kit receptor can interact with and phosphorylate the EPOR, contributing to its activation (Wu *et al*, 1995; Jacobs-Helber, Penta *et al*, 1997; Wu *et al*, 1997). Signaling from the EPOR phosphorylates and stabilizes the thioredoxin-interacting protein, which promotes c-Kit expression (Held, Greenfest-Allen *et al*, 2020). Signaling by EPOR/STAT5 activates PIM1 gene expression (Sathyanarayana, Dev *et al*, 2008; Gillinder *et al*, 2017), which may potentiate MAPK/ERK signaling (Wang *et al*, 2012), a possibility that is supported by our finding that *PIM1* suppression inhibits erythroblast proliferation. The transcription factor GATA2, expressed during early erythropoiesis, promotes *Kit* gene transcription directly, and also indirectly by stimulating the expression of *Samd14* (Hewitt, Kim *et al*, 2015; Hewitt, Katsumura *et al*, 2017), which enhances c-Kit signaling. The erythroid transcription factor GATA1, which drives the expression of terminal maturation markers, also inhibits transcription of the *Gata2* gene (Grass, Jing *et al*, 2006; Bresnick, Lee *et al*, 2010) and *Kit* gene (Vitelli, Condorelli *et al*, 2000; Jing, Vakoc *et al*, 2008). Additionally, GATA1 inhibits the expression of multiple subunits of the exosome complex, which promotes c-Kit expression (McIver, Katsumura *et al*, 2016). Our transcriptome and proteomic datasets recapitulate the transition from c-Kit dominant Ras/MAPK/ERK activity to EPOR/Stat5 activity during erythropoiesis and provide a unique, developmental stage-specific dataset of protein, phosphoprotein, and mRNA expression to explore this process further.

Phosphoproteomics technology is still rapidly improving and we expect that it will soon give us a virtually complete set of all the functional sites that have been described in the literature over many years (such as all pertinent phosphosites on EPOR and some MAPK-ERK components). Our studies were performed on erythroblasts cultured *in vitro* under defined concentrations of cytokines, nutrients, and oxygen. Thus, it will be interesting and informative to compare our results to findings obtained from bone marrow-derived erythroblasts. Furthermore, given the unexpectedly large role of phosphosignaling during erythropoiesis defined by our unbiased global, high-resolution proteomic study, it will now be interesting to investigate other post-translational protein modifications, which

could employ different enrichment steps but similar strategies for bioinformatic and functional follow-up. In this regard, we already observed distinct regulation of more than a hundred members of the ubiquitin machinery, making this post-translational modification particularly exciting for further explorations.

# Materials and Methods

### CD34$^+$ cell culture and manipulation

CD34$^+$ cells were obtained from de-identified healthy donors (Key Biologics, Lifeblood). All healthy donors provided informed consent. Human subject research protocols are approved by local ethical committees: St. Jude Children's Research Hospital protocol "Bone marrow for hemoglobinopathy research" (NCT00669305). Briefly, CD34$^+$ hematopoietic stem and progenitor cells (HSPCs) were mobilized from normal subjects by granulocyte colony-stimulating factor, collected by apheresis, and enriched by immunomagnetic bead selection using an autoMACS Pro Separator (Miltenyi Biotec), according to the manufacturer's protocol. At least 95% purity was achieved, as assessed by flow cytometry using a PE-conjugated anti-human CD34 antibody (Miltenyi Biotec, clone AC136, #130-081-002). A 3-phase culture protocol was used to promote erythroid differentiation and maturation. In phase 1 (days 0–7), cells were cultured at a density of $10^5$–$10^6$ cells/ml in IMDM with 2% human AB plasma, 3% human AB serum, 1% penicillin/streptomycin, 3 IU/ml heparin, 10 μg/ml insulin, 200 μg/ml holo-transferrin, 1 IU EPO, 10 ng/ml SCF, and 1 ng/ml IL-3. In phase 2 (days 8–12), IL-3 was omitted from the medium. In phase 3 (days 12–18), cells were cultured at a density of $10^6$/ml, with both IL-3 and SCF being omitted from the medium and the holo-transferrin concentration increased to 1 mg/ml. Erythroid differentiation and maturation were monitored by flow cytometry, using FITC-conjugated anti-CD235a (BD Biosciences, clone GA-R2, #561017), APC-conjugated anti-Band3 (gift from Xiuli An Lab in New York Blood Center), and VioBlue-conjugated anti-CD49d (Miltenyi, clone MZ18-24A9, #130-099-680).

### CRISPR/Cas9 screen with kinase-domain library

We utilized a kinase domain-focused sgRNA library targeting 482 kinase domains, which covers almost all annotated kinases in the human genome, and 50 non-targeting sgRNAs as negative controls (Addgene #117725) (Manning, Whyte *et al*, 2002; Grevet *et al*, 2018; Tarumoto, Lu *et al*, 2018a). Six independent sgRNAs were designed for targeting domain regions of each gene. All sgRNAs were designed using an algorithm to exclude those with high off-target effects (Hsu, Scott *et al*, 2013). Domain targeting and positive/negative control sgRNAs were synthesized in duplicate or triplicate in a pooled format on an array platform (Twist Bioscience) and then PCR-cloned into the BsmB1 site of the lentiviral vector LRG2.1 (Addgene: #108098) using a Gibson Assembly kit (NEB).

Approximately $12 \times 10^6$ HUDEP-2 cells stably expressing Cas9 were incubated with the sgRNA library at multiplicity of infection (MOI) of ~0.3 to achieve approximately 40% GFP positive cells, providing an approximately 1,000-fold library coverage, with most transduced cells containing a single copy of the vector. Two days

after infection, GFP$^+$ cells were purified by FACS and then maintained in expansion medium for 6 days (total of 8 days post-transduction). Half of the cells were then maintained in expansion culture for an additional 10 days and the other half were transitioned to a differentiation medium for 3 days to induce hemoglobin synthesis. Erythroid maturation was monitored by flow cytometry, using FITC-conjugated anti-CD235a (BD Biosciences, clone GA-R2), APC-conjugated anti-Band 3 (gift from Xiuli An Lab in New York Blood Center), and Violet Blue-conjugated anti-CD49d (Miltenyi, clone MZ18-24A9). Band3$^+$ and Band3$^-$ cell populations from the CD235a$^+$ cell fraction were purified by FACS. The representation of each lentiviral vector-encoded sgRNA was compared in the two cell populations by next-generation sequencing.

Library preparation and next-generation sequencing were performed as described (Grevet et al, 2018; Tarumoto et al, 2018a). Genomic DNA was extracted using the DNeasy Blood and Tissue kit (Qiagen). Reactions were performed with 24 cycles of amplification with 200 ng of gDNA in 25 μl CloneAmp enzyme system (Takara Bio). Eight parallel reactions were performed to maintain sgRNA library representation. PCRs were then pooled for each sample and purified with the QIAGEN PCR purification kit. PCR products were analyzed on an agarose gel, and the DNA band of expected size was excised and purified. MiSeq 250-bp paired-end sequencing (Illumina) was performed. For data analysis, FastQ files obtained after MiSeq sequencing were demultiplexed using the MiSeq Reporter software (Illumina). Paired reads were trimmed and filtered using the CLC Genomics Workbench (Qiagen) and matched against sgRNA sequences within the library. Read counts for each sgRNA were normalized against total read counts across all samples. The MAGeCK methods were used for differential analysis for sgRNA and gene rankings (Li, Xu et al, 2014; Wang, Wang et al, 2019). $P < 0.05$ was considered to be statistically significant.

## HUDEP-2 cell culture and induced maturation

Mycoplasma-free HUDEP-2 cells were cultured as described (Kurita et al, 2013). Immature cells were expanded in the StemSpan serum-free medium (SFEM; Stem Cell Technologies) supplemented with 1 μM dexamethasone, 1 μg/ml doxycycline, 50 ng/ml human stem cell factor (SCF), 3 units/ml erythropoietin (EPO), and 1% penicillin–streptomycin. To induce erythroid maturation, HUDEP-2 cells were cultured in a differentiation medium composed of IMDM base medium (Invitrogen) supplemented with 2% FBS, 3% human serum albumin, 3 units/ml EPO, 10 μg/ml insulin, 1,000 μg/ml holo-transferrin, and 3 units/ml heparin. Erythroid differentiation and maturation were monitored by flow cytometry, using FITC-conjugated anti-CD235a (BD Biosciences, clone GA-R2, #561017), APC-conjugated anti-Band3 (gift from Xiuli An Lab in New York Blood Center), and VioBlue-conjugated anti-CD49d (Miltenyi, clone MZ18-24A9, #130-099-680).

## CRISPR/Cas9-mediated genome editing of HUDEP-2 cells

Individual sgRNAs identified by library screening were generated as complementary oligonucleotides, annealed, and cloned into the BsmBI site of the pXPR_003 lentiviral vector. Vector particles generated from 293T cells were used to transduce HUDEP-2 cells stably expressing Cas9. Cells were incubated for 7–10 days with 10 μg/ml blasticidin and 1 μg/ml puromycin to select for transduction with sgRNA and Cas9 vectors, respectively.

To characterize on-target indel mutations, DNA was extracted from genome edited cells with the DNeasy Blood and Tissue kit (Qiagen). The first round PCR amplification using CloneAmp HiFi PCR Premix (Takara) was performed with locus-specific primers containing universal 5' nextera-adaptor sequence (Table EV4). The amplicon library was prepared with indexing primers (P5-dual-index-F and P7-dual-index-R), and followed by sequencing on the Illumina MiSeq platform with $2 \times 150$-bp paired-end reads at the Hartwell Sequencing Center (St. Jude Core facility). Sequence alignment and mutation detection were performed using CRISPResso2 software (Clement, Rees et al, 2019).

## Cell lysates and immunoblot analysis

Cells were suspended in Thermo Scientific Pierce IP Lysis Buffer (Thermo Fisher #87787) supplemented with 1 mM phenylmethylsulfonyl fluoride, and 1:500 protease inhibitor cocktail (Sigma-Aldrich). Proteins were resolved on polyacrylamide gels (Bio-Rad), transferred to a PVDF membrane, and incubated in blocking buffer (5% milk in TBST). Antibody staining was visualized using the Odyssey CLx Imaging System.

## (Phospho)proteome sample preparation for MS analysis

All MS experiments were performed in biological quadruplicates. Cell pellets were lysed in SDC buffer (4% sodium deoxycholate in 100 mM Tris pH 8.5) and heated for 5 min at 95°C. Lysates were cooled on ice and sonicated. Protein concentration was determined by Tryptophan assay (Kulak, Pichler et al, 2014). We reduced disulfide bonds and carbamidomethylate cysteine residues by adding TCEP and 2-Chloroacetamide to the final volumes of 10 mM and 40 mM, respectively, for 5 min at 45°C. Proteins were subsequently digested by the addition of 1:100 LysC and Trypsin overnight at 37°C with agitation (1,500 rpm). Next day, 20 μg of protein material was aliquoted and processed using an in-StageTip (iST) protocol (Kulak et al, 2014). Approximately 500 ng peptide was used for single shot DIA analysis, while the rest (~10 ug) of clean peptides were fractionated using the high-pH reversed-phase "Spider fractionator" into 8 fractions to generate deep proteomes to build spectral library (Kulak, Geyer et al, 2017). 80 μg of peptides were used for phosphopeptide enrichment using the EasyPhos workflow as described previously (Humphrey et al, 2015; Humphrey et al, 2018). After mixing peptides with Isopropanol and EP enrichment buffer (48% TFA, 8 mM KH$_2$PO$_4$), they were enriched with 5 mg of TiO$_2$ beads which were prepared at a concentration of 1 mg/μl in loading buffer (6% TFA/80% ACN (vol/vol)) and incubated at 40°C with shaking (2,000 rpm) for 5 min. Afterward, the phosphopeptide-containing TiO$_2$ beads were further washed with 4 ml wash buffer (5% TFA/60% ISO (vol/vol)) and treated with elution buffer (40% ACN, 15% NH$_4$OH). Eluted phosphopeptides were concentrated in a SpeedVac for 20 min at 45°C and using StageTips loaded with SDB-RPS disks. 6 μl MS loading buffer (0.2% TFA/2% ACN (vol/vol)) was added to the dried samples prior to LC-MS/MS analysis.

## Liquid chromatography-MS analysis

Nanoflow LC-MS/MS measurements were carried out on an EASY-nLC 1200 system (Thermo Fisher Scientific) combined with the latest generation linear quadrupole Orbitrap instrument (Q Exactive HF-X) coupled to a nano-electrospray ion source (Thermo Fisher Scientific). We always used a 50 cm HPLC column (75 μm inner diameter, in-house packed into the tip with ReproSil-Pur C18-AQ 1.9 μm resin (Dr. Maisch GmbH)). Column temperature was kept at 60°C by a Peltier element containing in-house developed oven.

500 ng peptides were analyzed with a 100 min gradient. Peptides were loaded in buffer A (0.1% formic acid (FA) (v/v)) and eluted with a linear 80 min gradient of 5–30% of buffer B (80% acetonitrile (ACN) plus 0.1% FA (v/v)), followed by a 4 min increase to 60% of buffer B and a 4 min increase to 95% of buffer B, and a 4 min wash of 95% buffer B at a flow rate of 300 nl/min. Buffer B concentration was decreased to 4% in 4 min and stayed at 4% for 4 min.

For the analysis of the fractions, the instrument was operated in the DDA mode (Top12). The resolution of the Orbitrap analyzer was set to 60,000 and 15,000 for MS1 and MS2, with a maximum injection time of 20 ms and 60 ms, respectively. The mass range monitored in MS1 was set to 300–1,650 $m/z$. The automatic gain control (AGC) target was set to 3e6 and 1e5 in MS1 and MS2, respectively. The fragmentation was accomplished by higher energy collision dissociation at a normalized collision energy setting of 27%. Dynamic exclusion was 20 sec.

For single shot samples, the instrument was operated in the DIA mode. Every MS1 scan (350–1,650 $m/z$, 120,000 resolution at $m/z$ 200, AGC target of 3e6 and 60 ms injection time) was followed by 33 MS2 windows ranged from 300.5 $m/z$ (lower boundary of first window) to 1649.5 $m/z$ (upper boundary of 33rd window). This resulted in a cycle time of 3.4 s. MS2 settings were an ion target value of $3 \times 10^6$ charges for the precursor window with an Xcalibur-automated maximum injection time and a resolution of 30,000 at $m/z$ 200. The fragmentation was accomplished by higher energy collision dissociation with stepped collision energies of 25.5, 27 and 30%. The spectra were recorded in profile mode. The default charge state for the MS2 was set to 3. Data were acquired with Xcalibur 4.0.27.10 and Tune Plus version 2.1 (Thermo Fisher).

Phosphopeptides were analyzed with a 100 min gradient. Peptides were loaded in buffer A (0.1% formic acid (FA) (v/v)) and eluted with a linear 60 min gradient of 3–19 of buffer B (80% acetonitrile (ACN) plus 0.1% FA (v/v)), followed by a 30 min increase to 41% of buffer B and a 5 min increase to 90% of buffer B, and a 5 min wash of 90% buffer B at a flow rate of 350 nl/min. The instrument was operated in the DDA mode (Top10). The resolution of the Orbitrap analyzer was set to 60,000 and 15,000 for MS1 and MS2, with a maximum injection time of 120 and 50 ms, respectively. The mass range monitored in MS1 was set to 300–1,600 $m/z$. The automatic gain control (AGC) target was set to 3e6 and 1e5 in MS1 and MS2, respectively. The fragmentation was accomplished by higher energy collision dissociation at a normalized collision energy setting of 27%. Dynamic exclusion was 30 s.

## MS data analysis

The fractions (DDA) and the single shot samples (DIA) were used to generate a DDA-library and direct-DIA-library, respectively, which were combined into a hybrid library in Spectromine version 1.0.21621.8.15296 (Biognosys AG). The hybrid spectral library was subsequently used to search the MS data of the single shot samples in Spectronaut version 12.0.20491.9.26669 (Biognosys AG) for final protein identification and quantification. All searches were performed against the Human UniProt FASTA database (2017). Carbamidomethylation was set as fixed modification and acetylation of the protein N-terminus and oxidation of methionine as variable modifications. Trypsin/P proteolytic cleavage rule was used with a maximum of two miscleavages permitted and a peptide length of 7–52 amino acids. When generating the spectral library generation, minimum and maximum of number of fragments per peptide were set to 3 and 6, respectively. A protein and precursor FDR of 1% were used for filtering and subsequent reporting in samples (q-value mode).

For the phosphoproteome, raw MS data were processed using MaxQuant version 1.6.2.10 (Cox & Mann, 2008; Cox, Neuhauser et al, 2011) with an FDR < 0.01 at the peptide and protein level against the Human UniProt FASTA database (2017). Enzyme specificity was set to trypsin, and the search included cysteine carbamidomethylation as a fixed modification and N-acetylation of protein and oxidation of methionine and phosphorylation (SYT) as variable modifications. Up to two missed cleavages were allowed for protease digestion, and peptides had to be fully tryptic.

## Bioinformatics data analysis

We mainly performed data analysis in the Perseus (version 1.6.0.9) (Tyanova, Temu et al, 2016), Microsoft Excel and data visualized using GraphPad Prism (GraphPad Software) or RStudio (https://www.rstudio.com/). Apart from coefficient of variation, log2-transformed protein intensities were used for further analysis. Coefficients of variations were calculated for raw protein intensities between replicates individually. Phosphopeptides that were identified in the decoy reverse database were not considered for data analysis. Both data sets were filtered to make sure that identified proteins and phosphopeptides showed expression in all biological triplicates of at least one differentiation stage and the missing values were subsequently replaced by random numbers that were drawn from a normal distribution (width = 0.3 and down shift = 1.8). PCA of differentiation stages and biological replicates was performed as previously described in Deeb and Tyanova et al, (2015). Multi-sample test (ANOVA) for determining if any of the means of differentiation stages were significantly different from each other was applied to protein data set. For truncation, we used permutation-based FDR which was set to 0.05 in conjunction with an S0-parameter of 0.1. For hierarchical clustering of significant proteins, median protein abundances of biological replicates were z-scored and clustered using Euclidean as a distance measure for row clustering. Gene Ontology (GO) enrichments in the clusters were calculated by Fisher's exact test using Benjamini–Hochberg false discovery rate for truncation, setting a value of 0.02 as threshold. Mean log2 ratios of biological triplicates and the corresponding P-values were visualized with volcano plots. We chose a significance cutoff based on a FDR < 0.05 or 0.01 in volcano plots.

## Copy number calculation

Intensities were converted to copy number estimations using the proteomic ruler (Wisniewski et al, 2014). The proteomic ruler plugin

v.0.1.6 was downloaded from the Perseus plugin store, for use with Perseus version 1.5.5.0. Protein intensities were filtered for 100% data completeness in at least one stage. Protein groups were annotated with amino acid sequence and tryptic peptide information for the leading protein ID, using the FASTA file used for processing data. Copy numbers were estimated using the following settings: averaging mode—"All columns separately", molecular masses—"average molecular mass", scaling mode—"Histone proteomic ruler", ploidy "2", and total cellular protein concentration—"200 g/l".

### In silico deconvolution approach

All marker sets (except of "any20") were filtered for an ANOVA *q*-value < 0.01. For the cluster markers, the top three most significant (smallest ANOVA *q*-value) proteins for each cluster were selected. The combined markers contain all proteins from the sorting markers, known markers, cluster markers, and SLC markers. The "any20" markers were selected by randomly picking 20 proteins from the unfiltered protein list, excluding proteins included in any of the other marker lists.

A signature matrix was generated for each of the six marker sets. Only two out of the four cell type replicates, replicates 2 and 4, were used for generating the signature matrices containing averaged, non-logged intensity values of each marker protein. The other two replicates (replicates 1 and 3) were subsequently used for creating in silico mixture populations.

The ratios for mixing the different cell types were determined by randomly picking 500 combinations of 5 values (corresponding to the 5 cell types) that add up to 1:

$$\sum_{i=1}^{5} a_i = 1 \qquad (1)$$

The mixture intensity $I_{\mathrm{mix}}$ of each protein $k$ was then determined by summing up the intensity of protein $k$ in cell type $i$, $I_i^k$, multiplied by the fraction of cells from cell type $a_i$.

$$I_{\mathrm{mix}}^k = \sum_{i=1}^{5} I_i^k a_i \qquad (2)$$

Next, using the mixture intensities $I_{\mathrm{mix}}$ of each marker protein $k$ (equation 2) as well as the signature matrix, we set out to estimate the fractions of cells contributed by each of the five cell types $\dot{a}_i$. The closer the estimations $\dot{a}_i$ are to the true mixing ratios $a_i$, the better the marker set is for deconvoluting and differentiating the five cell types.

Writing equation 2 in matrix form when several proteins are evaluated at the same time, the model expands to the following:

$$\underline{i}_{\mathrm{mix}} = I\underline{\dot{a}} \qquad (3)$$

Here, $\underline{i}_{\mathrm{mix}}$ is a vector of mixture intensities for each evaluated marker protein $k$, $I$ is the signature matrix containing the intensities of each evaluated marker protein $k$ (rows) in each cell type $i$ (columns), and $\underline{\dot{a}}$ is the vector with the fractions of cells in each cell type $i$ that we aim to estimate.

To estimate $\underline{\dot{a}}$, we can solve the linear equation system (equation 3) by using a minimum least squares optimization (python

scipy.optimize.minimize). Boundary conditions set the minimum possible values of $\underline{\dot{a}}$ to zero in order to avoid negative values. After an initial estimation of $\underline{\dot{a}}$, only the top 90th percentile of marker proteins for which the estimates fit best are kept for solving the linear equation system (equation 3) in a second iteration. The estimated vector of fractions $\underline{\dot{a}}$, is finally normalized by the Manhattan (L1) norm.

To evaluate the results of the deconvolution step, we implemented a weighted error metric to estimate how close the estimated ratios $\underline{\dot{a}}$ are to the true ratios $\underline{a}$. Here, mistakes in assigning neighboring cell types (e.g., between P3 and P4) contribute less to the overall error than mistakes between far distant cell types (e.g., Progenitors and P5). In addition to evaluating the six sets of protein markers, three controls were generated: "random" uses a random ratio estimation; "uniform" assumes a uniform ratio estimation ($\dot{a}_1 = 0.2$); and "center" assumes that all cells are of type P3 ($\dot{a}_3 = 1$).

## Data availability

All MS proteomics data have been deposited on ProteomeXchange via the PRIDE database with the dataset identifier PXD017276 (https://www.ebi.ac.uk/pride/archive/projects/PXD017276). All other data supporting findings of this study are available within this article and in Expanded View tables and datasets.

**Expanded View** for this article is available online.

### Acknowledgements

This work was supported by the Max Planck Society for the Advancement of Science and by the Deutsche Forschungsgemeinschaft (DFG, German Research Foundation)—SCHU 3196/1-1. We thank Florian Meier, Igor Paron, Christian Deiml, Philipp Geyer, Johannes B Mueller, Fynn M Hansen, Sebastian Virreira Winter, and all the members of the departments of Proteomics and Signal Transduction and Molecular Machines and Signaling at Max Planck Institute of Biochemistry for their assistances and helpful discussions. We also thank the NCI Cancer Center grant to St. Jude (NIHP30CA021765) and St Jude Core facilities including Flow Core for cell sorting, Hartwell Center for NGS, and Image Core for Cytospin Scanning. Open Access funding enabled and organized by Projekt DEAL.

### Author contributions

OK performed proteomics experiments and analyzed the data. PX and YY performed FACS sorting, tissue culture experiments and CRISPR/Cas9 screen and biological validation assay for PIM1. IB developed the bioinformatics deconvolution approach to validate markers. ARFC and AS helped with the bioinformatics analysis of phosphoproteome data. SVB helped with the analysis and the interpretation of the data. OK, PX, AFA, SVB, BAS, MW, and MM designed the study and wrote the paper. BAS, AFA, MW, and MM coordinated and supervised.

### Conflict of interest

The authors declare that they have no conflict of interest.

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
