## [Review Process File · Molecular Systems Biology]

Integrative proteomics reveals principles of dynamic phospho-signaling networks in erythropoiesis

Matthias Mann, Ozge Karayel, Peng Xu, Isabell Bludau, Senthil Bhoopalan, Yu Yao, Ana Rita Freitas Colaco, Alberto Santos, Brenda Schulman, Arno Alpi, and Mitchell Weiss

DOI: 10.15252/msb.20209813

Corresponding author(s): Matthias Mann (mmann@biochem.mpg.de), Brenda Schulman (schulman@biochem.mpg.de), Arno Alpi (aalpi@biochem.mpg.de), Mitchell Weiss (Mitch.Weiss@STJUDE.ORG)

Review Timeline:

Submission Date:	25th Jun 20
Editorial Decision:	13th Aug 20
Revision Received:	9th Oct 20
Editorial Decision:	16th Oct 20
Revision Received:	23rd Oct 20
Accepted:	26th Oct 20

Editor: Maria Polychronidou

Transaction Report:

Thank you again for submitting your work to Molecular Systems Biology. We have now heard back from the three referees who agreed to evaluate your study. Overall, the reviewers are supportive. However, they raise a series of concerns, which we would ask you to address in a revision.

I think that the issues raised by the reviewers are clear and seem relatively straightforward to address. Please let me know in case you would like to discuss in further detail any of the issues raised, I would of course be happy to do that.

On a more editorial level, we would ask you to address the following issues:

REFeree REPORTS

Reviewer #1:

The manuscript by Karayel et al. describes an extensive proteomic/phosphoproteomic analysis during human erythroid differentiation. Several groups have provided rigorous data on the mammalian erythroid cell proteome, and this study yielded datasets that considerably extend this prior work. One outcome was the identification of differentially expressed transmembrane proteins that are of considerable interest in the field. Another outcome involves detailed information on kinases involved in differentiation. Overall, the data was rigorously generated/analyzed, the manuscript harbors important findings, and the datasets will catalyze further discoveries. Specific

recommendations are provided to further strengthen the manuscript.

Specific Comments:

- 1) Page 10, "optimized digestion and lysis protocol" This has potential to be critical and deserves fleshing out: what specific conditions does this statement refer to that increase transmembrane protein recovery and detection?
- 2) Page 16, legend - "downstreat"
- 3) The authors focused on transmembrane proteins and specifically SLCs, and they suggest the differential expression of SLCs reflects major differences in metabolic requirements during erythroid differentiation. It would be informative to parse the transmembrane proteins into SLCs and non-SLCs and assess whether a particular group has unique expression attributes relative to the other, or if the non-SLC cohort mimics the SLC cohort e.g. at certain differentiation transitions.
- 4) Multiple studies have implicated Ras/MAPK-dependent mechanisms in erythroid differentiation including early work from Zhang and Lodish (Blood 2004), as well as from Wojchowski (Pircher, JBC 2001; Held, Expt Hem 2020). It would help to more effectively describe what was known prior to the current work and how the new work advances knowledge on this problem.
- 5) Page 21, legend - "targetign"
- 6) Page 22 - Kit regulation during erythroid differentiation has been pretty extensively studied, and the authors conclude that "downregulation of c-Kit/MAPK signaling is a key driver of maturation". As discussed, it was known that GATA1 represses Kit expression, and in work not discussed, from the Bresnick group, the exosome complex sustains c-Kit expression and signaling, preventing differentiation (McIver, Elife 2016) and SAMD14 enhances c-Kit signaling, favoring the undifferentiated state (Hewitt, Mol Cell 2015; Dev Cell 2017). As noted above for Ras/MAPK, a more thorough discussion of what was known would facilitate comprehension vis-à-vis how the current elegant analysis advances or transforms prior knowledge.
- 7) CRISPR screen - Rather than referring to the prior papers, it would be ideal to comprehensively describe the design strategy to ensure reproducibility. The execution of the screen was well-described.
- 8) HUDEP-2 cell culture - These cells exhibit variable growth and differentiation properties in diverse labs and even within a lab. It would be important to provide exact details on where all reagents were obtained. Most of this information was not stated.
- 9) Validation of genome editing - It was stated that validation involved "next-gen sequencing or TIDE-seq analysis from Sanger sequencing datasets". Given the critical importance of validation, and the complex considerations associated with generation of many mutants, it would be important to expand this discussion of validation.
- 10) Page 29, "peptideswere"

Reviewer #2:

Karayel et al., report their results on mass spectrometry (MS)-based proteomics analysis of human cultured CD34+-derived erythroid cells at distinct stages of maturation. They observe dynamic changes in the proteome. Employing a CRISPR/Cas9 screen targeting kinases during erythroid maturation of HUDEP2 cultured line, they find targeting c-Kit/MAPK signaling promotes terminal erythroid maturation. The authors have done massive work and the functional approaches are sound. Exposing proteomic dynamic changes during human erythroid maturation is of major interest. MAPK is a known regulator of erythroid proliferation that blocks maturation in both mouse and human erythroid cells (literature on this includes PMID: 15705783; 17317860; PMID: 15166036; PMID: 12969966; PMID: 31413092 ; PMID: 15030167). The authors should reference these, and either clarify the novelty of their finding on MAPK in erythroid cell maturation or discuss the confirmatory aspect of their work in the context of what is known. In addition, it would be informative if the authors could discuss their results in the context of erythropoiesis of knock out mouse models of MAPK signaling proteins. Similarly, the authors should reference the work on PIM1 kinase induced by erythropoietin receptor signaling (PMID: 28732065, PMID: 20639905) and discuss their work in that context.

The authors identify a large number of SLC (solute carrier) transporters expressed during erythroid maturation. This is an interesting finding confirming previous transcriptomic analyses (that should also be referenced). It would have been interesting for the authors to provide functional information on SLCs in human erythroid cells.

Specific comments:

Figure 5 data on MAPK, HUDEP-2 cells have been cultured for 11 days. The authors should provide information regarding intermediary stages of expansion and cell death during culture. How have they excluded off-target effects?

Please clarify: "Glucose uptake during maturation appeared to roughly track with EPOR expression, reaching a maximal value when EPO response was highest, perhaps because of regulation by EPO stimulation in erythroid progenitor cells, as reported previously (Rogers et al., 2010)." And" In line with the growing need for glucose during maturation, two out of four identified SLCs transporting glucose (SLC2A1 and SLC2A4) gradually increased from progenitors to Ortho".

There seems to be a contradiction in the statement regarding erythropoietin receptor signaling during maturation (which is highest in mature erythroid progenitor/immature precursors) and the need for glucose that is stated to increase gradually to the Ortho stage, that needs to be addressed.

Reviewer #3:

Karayel et al. compiled a detailed and comprehensive description of the continuously changing protein expression and protein phosphorylation in sequential stages of human erythropoiesis, using an in-vitro differentiation model of primary erythroid cells. Their work provides a resource and a reference for researchers in the field, and also draws, in broad strokes, some key principles of the process of erythroid terminal differentiation. The sheer quantitative scale of their finding is in itself of interest: they find that thousands of proteins are both expressed and are being dynamical

regulated during terminal differentiation. It may encourage similar work in other differentiation models.

Although much of what they describe is not entirely new, the sheer scale, detail and depth of their analysis puts previous work in a larger and rigorous context. As example, some of the first solute carriers and transporters were cloned and characterized in red cells, notably, Band3 (Slc4a1) and glucose transporter. Here the authors document at least 68 solute carriers whose expression changes significantly during the process of differentiation. They document distinct protein signatures to each precursor stage. They provide rigorous confirmation to previous reports that histones decrease during late erythropoiesis prior to enucleation. They further assess the entire kinome, and carry out a complementary study in the phosphoproteome, together allowing a comprehensive documentation of the kinases and their substrates during each precursor stage. Further, they use a CRISPR/Cas9 screen to identify functionally the relevant kinases regulating erythroid differentiation in HUDEP-2 cells.

There are a few issues that the authors could address:

1. Readability. The authors use jargon that would not necessarily be comprehensible outside the proteomics or computational fields without much explanation. They should add a simple explanation to provide the reader with at least an intuition as to what is being discussed. E.g. What are DIA and DDA, what is an m/z precursor window. The second paragraph on page 12 is not comprehensible.
2. Throughout their work, the authors find that the largest changes in the proteome, phosphoproteome and kinome are at the beginning (progenitor to ProE) and end (Poly to ortho) of terminal differentiation. The changes described at the beginning tie well with previous literature that suggests this transition is a sharp switch both transcriptionally and at the level of chromatin. It includes sudden changes in the cell cycle, in growth factor responsiveness (this is the time when Epo dependence begins) and transcription (it is at this transition that Gata1 targets such as globins begin to be transcribed rapidly). It would be good for the authors to put their findings in this context. Further, the dramatic proteome changes at the end of terminal differentiation (Poly to Ortho) suggests an abrupt regulatory event, not to my knowledge documented in the literature- it would be good for the authors to highlight that.
3. The authors might want to highlight the interesting finding that DNA damage checkpoint kinases including ATM, ATR and Check2 are enriched in erythroid precursors. It is clear evidence against the common belief that erythroblasts would not need to 'invest' in DNA repair because of impending enucleation.
4. The authors conclude that both Kit and MAPK pathways are downregulated early, whereas EpoR and Stat5 persist into later differentiation stages. They then draw the conclusion that Kit signals primarily through MAPK, whereas EpoR signals through Stat5. This conclusion is not justified simply based on data that would be correlative at best. Further, EpoR signaling through MAPK is well documented. The authors interpret the data in Figure 6D to suggest that EpoR remains uniformly high but it too seems to decline immediately following its peak in ProE, a finding consistent with previous reports using radioligand binding. Finally, the time course of the various MAPKs is more complex than the simple decline seen with Kit.

Point-by-point answers to ‘Integrative proteomics reveals principles of dynamic phospho-signaling networks in human erythropoiesis’ by Karayel and Xu et al.

We are delighted that the reviewers found our paper of considerable interest and we thank them for their evaluation and constructive comments.

Reviewer #1:

The manuscript by Karayel et al. describes an extensive proteomic/phosphoproteomic analysis during human erythroid differentiation. Several groups have provided rigorous data on the mammalian erythroid cell proteome, and this study yielded datasets that considerably extend this prior work. One outcome was the identification of differentially expressed transmembrane proteins that are of considerable interest in the field. Another outcome involves detailed information on kinases involved in differentiation. Overall, the data was rigorously generated/analyzed, the manuscript harbors important findings, and the datasets will catalyze further discoveries. Specific recommendations are provided to further strengthen the manuscript.

We thank the reviewer for their positive and constructive feedback.

Specific Comments:

1) Page 10, "optimized digestion and lysis protocol" This has potential to be critical and deserves fleshing out: what specific conditions does this statement refer to that increase transmembrane protein recovery and detection?

We used sodium deoxycholate (SDC), an ionic and denaturant surfactant, for both cell lysis and protein digestion. Detergents such as SDC are preferred for cell lysis as they denature proteins (breaking protein-protein interactions) as well as provide efficient solubilization of hydrophobic proteins or membranes. SDC also can be also used for digestion of protein and in fact it enhances trypsin activity many-folds at low concentrations (~1%) (Leon et al., MCP, 2013), allowing to perform the whole sample preparation steps in one buffer system. It can also be easily removed during peptide clean-up procedure using SDB-RPS StageTips (Kulak et al., Nature Methods, 2014). Deoxycholate-assisted in-solution digestion combined with SDB-RPS clean-up allows for efficient, unbiased generation and recovery of peptides from all protein classes, including membrane proteins and especially makes the sample preparation workflows for proteomics more effective, high-throughput, and reproducible. We have now

included a short explanation for our 'optimized digestion and lysis protocol' in the manuscript (Page 8, Line 209-212) and a detailed protocol is provided in the Method section.

2) Page 16, legend - "downstreat"

Thank you for spotting these errors! We have corrected them in the text.

3) The authors focused on transmembrane proteins and specifically SLCs, and they suggest the differential expression of SLCs reflects major differences in metabolic requirements during erythroid differentiation. It would be informative to parse the transmembrane proteins into SLCs and non-SLCs and assess whether a particular group has unique expression attributes relative to the other, or if the non-SLC cohort mimics the SLC cohort e.g. at certain differentiation transitions.

Across the differentiation stages, we found 692 membrane proteins that changed significantly (ANOVA FDR<0.01, EV Table 1). Of the significantly changing membrane proteins, an overrepresentation analysis identified proteins that transport small molecules across membranes as one of the most represented (p 8.7 E-09). Further functional classification showed strong enrichment of SLC proteins (Figure 3A). Given the central importance of SLC proteins in erythropoiesis, we primarily focused on this class of transmembrane proteins. Although the non-SLC proteins largely mimic the dynamic changes of the SLC cohort in abundance, we observed that a few other protein classes with transporter function exhibit unique expression characteristics, for instance, the ABC-family proteins. In fact, our unbiased analysis already revealed the enrichment of this particular protein family (Figure 3A). Our data quantified six members of the ABC-family proteins that significantly change in at least one transition (ANOVA FDR<0.01). They all showed dynamic upregulation during the maturation stages. We now added this results as a panel to the EV Figure 4.

4) Multiple studies have implicated Ras/MAPK-dependent mechanisms in erythroid differentiation including early work from Zhang and Lodish (Blood 2004), as well as from Wojchowski (Pircher, JBC 2001; Held, Expt Hem 2020). It would help to more effectively describe what was known prior to the current work and how the new work advances knowledge on this problem.

Thank you for pointing this out. We added a more complete review of this topic, including relevant reference citations to the Discussion (Lines 566-604, Page 18-19).

5) Page 21, legend - "targetign"

Corrected.

6) Page 22 - Kit regulation during erythroid differentiation has been pretty extensively studied, and the authors conclude that "downregulation of c-Kit/MAPK signaling is a key driver of maturation". As discussed, it was known that GATA1 represses Kit expression, and in work not discussed, from the Bresnick group, the exosome complex sustains c-Kit expression and signaling, preventing differentiation (McIver, Elife 2016) and SAMD14 enhances c-Kit signaling, favoring the undifferentiated state (Hewitt, Mol Cell 2015; Dev Cell 2017). As noted above for Ras/MAPK, a more thorough discussion of what was

known would facilitate comprehension vis-à-vis how the current elegant analysis advances or transforms prior knowledge.

The revised Discussion reviews this work and includes the relevant reference citations (Lines 558-604, Page 18).

7) CRISPR screen - Rather than referring to the prior papers, it would be ideal to comprehensively describe the design strategy to ensure reproducibility. The execution of the screen was well-described.

We added additional details and clarification to the protocol to the Methods section on page 25.

8) HUDEP-2 cell culture - These cells exhibit variable growth and differentiation properties in diverse labs and even within a lab. It would be important to provide exact details on where all reagents were obtained. Most of this information was not stated.

These details are provided in new EV Table 6.

9) Validation of genome editing - It was stated that validation involved "next-gen sequencing or TIDE-seq analysis from Sanger sequencing datasets". Given the critical importance of validation, and the complex considerations associated with generation of many mutants, it would be important to expand this discussion of validation.

We added additional details on indel characterization after genome editing to the Methods section on page 26 and provide the indel frequencies in EV Figure 6A and 6C. The first-round, gene-specific PCR primers used to characterize indel frequencies are described in new EV Table 6.

10) Page 29, "peptideswere"

Corrected.

Reviewer #2:

Karayel et al., report their results on mass spectrometry (MS)-based proteomics analysis of human cultured CD34+-derived erythroid cells at distinct stages of maturation. They observe dynamic changes in the proteome. Employing a CRISPR/Cas9 screen targeting kinases during erythroid maturation of HUDEP2 cultured line, they find targeting c-Kit/MAPK signaling promotes terminal erythroid maturation. The authors have done massive work and the functional approaches are sound. Exposing proteomic dynamic changes during human erythroid maturation is of major interest.

We thank the reviewer for their detailed and positive comments.

MAPK is a known regulator of erythroid proliferation that blocks maturation in both mouse and human erythroid cells (literature on this includes PMID: 15705783; 17317860; PMID: 15166036; PMID: 12969966; PMID: 31413092 ; PMID: 15030167). The authors should reference these, and either clarify the novelty of their finding on MAPK in erythroid cell maturation or discuss the confirmatory aspect of their work in the context of what is known. In addition, it would be informative if the authors could discuss their results in the context of erythropoiesis of knock out mouse models of MAPK signaling proteins. Similarly, the authors should reference the work on PIM1 kinase induced by erythropoietin receptor signaling (PMID: 28732065, PMID: 20639905) and discuss their work in that context. The authors identify a large number of SLC (solute carrier) transporters expressed during erythroid maturation. This is an interesting finding confirming previous transcriptomic analyses (that should also be referenced). It would have been interesting for the authors to provide functional information on SLCs in human erythroid cells.

These points and citations are now included in the Discussion on pages 18-19. We thank the reviewer for providing us with relevant PMID numbers of citations that allow us to improve the discussion of our findings.

Specific comments:

Figure 5 data on MAPK, HUDEP-2 cells have been cultured for 11 days. The authors should provide information regarding intermediary stages of expansion and cell death during culture. How have they excluded off-target effects?

Expansion of cells grown in expansion medium during intermediary stages is now shown in EV Figure 6B and D. Off-target effects of individual sgRNAs were not assessed directly, although we used a bioinformatic algorithm designed to minimize off target effects (now noted on Page 25). We acknowledge this as a limitation of our study on page 14 (lines 424-425).

Please clarify: "Glucose uptake during maturation appeared to roughly track with EPOR expression, reaching a maximal value when EPO response was highest, perhaps because of regulation by EPO stimulation in erythroid progenitor cells, as reported previously (Rogers et al., 2010)." And" In line with the growing need for glucose during maturation, two out of four identified SLCs transporting glucose (SLC2A1 and SLC2A4) gradually increased from progenitors to Ortho".

There seems to be a contradiction in the statement regarding erythropoietin receptor signaling during maturation (which is highest in mature erythroid progenitor/immature precursors) and the need for glucose that is stated to increase gradually to the Ortho stage, that needs to be addressed.

We thank the reviewer for spotting this apparent contradiction. We re-assessed the literature on glucose transporters in erythropoiesis (and excluded the references only referring to two meeting abstracts, Rogers et al. 2010 and Justus et al. 2019). We edited the paragraph (Page 9, Line 242-258) to highlight 1) the gradual increase of SLC2A1/GLUT1 concordant with increased uptake of L-dehydroascorbic acid, 2) the notable increase of SLC2A4/GLUT4 during terminal maturation. SLC2A4/GLUT4 is primarily

described in skeletal muscle and playing a key role in regulating blood glucose concentration, but the common profile with SLC2A1/GLUT1 indicating a contribution of SLC2A4/GLUT4 in glucose uptake in erythrocytes.

Reviewer #3:

Karayel et al. compiled a detailed and comprehensive description of the continuously changing protein expression and protein phosphorylation in sequential stages of human erythropoiesis, using an in-vitro differentiation model of primary erythroid cells. Their work provides a resource and a reference for researchers in the field, and also draws, in broad strokes, some key principles of the process of erythroid terminal differentiation. The sheer quantitative scale of their finding is in itself of interest: they find that thousands of proteins are both expressed and are being dynamically regulated during terminal differentiation. It may encourage similar work in other differentiation models.

Although much of what they describe is not entirely new, the sheer scale, detail and depth of their analysis puts previous work in a larger and rigorous context. As example, some of the first solute carriers and transporters were cloned and characterized in red cells, notably, Band3 (Slc4a1) and glucose transporter. Here the authors document at least 68 solute carriers whose expression changes significantly during the process of differentiation. They document distinct protein signatures to each precursor stage. They provide rigorous confirmation to previous reports that histones decrease during late erythropoiesis prior to enucleation. They further assess the entire kinome, and carry out a complementary study in the phosphoproteome, together allowing a comprehensive documentation of the kinases and their substrates during each precursor stage. Further, they use a CRISPR/Cas9 screen to identify functionally the relevant kinases regulating erythroid differentiation in HUDEP-2 cells.

We thank the reviewer for accurate description of our work and the generally positive evaluation. We found the comments very helpful in improving our manuscript.

There are a few issues that the authors could address:

1. Readability. The authors use jargon that would not necessarily be comprehensible outside the proteomics or computational fields without much explanation. They should add a simple explanation to provide the reader with at least an intuition as to what is being discussed. E.g. What are DIA and DDA, what is an m/z precursor window. The second paragraph on page 12 is not comprehensible.

Thank you for that comment and we are sorry if explanations for proteomics methods were insufficient. We have now explained DIA and DDA methods as well as the deconvolution approach in detail in the manuscript, mainly in the introduction (Page 4, Line 84-94) and results (Page 10, Line 283-301). Our method section also includes very detailed protocols for both methods.

2. Throughout their work, the authors find that the largest changes in the proteome, phosphoproteome and kinome are at the beginning (progenitor to ProE) and end (Poly to ortho) of terminal differentiation. The changes described at the beginning tie well with previous literature that suggests this transition is a sharp switch both transcriptionally and at the level of chromatin. It includes sudden changes in the cell cycle, in

growth factor responsiveness (this is the time when Epo dependence begins) and transcription (it is at this transition that Gata1 targets such as globins begin to be transcribed rapidly). It would be good for the authors to put their findings in this context. Further, the dramatic proteome changes at the end of terminal differentiation (Poly to Ortho) suggests an abrupt regulatory event, not to my knowledge documented in the literature- it would be good for the authors to highlight that.

We addressed this comment by describing and discussing the dramatic changes in the global proteome, phosphoproteome, and kinome (particularly observed from progenitor to ProE and Poly to ortho stage transition) in relevant paragraphs of the result section as well as in a paragraph of the revised discussion (Page 16-17, Lines 504-522).

3. The authors might want to highlight the interesting finding that DNA damage checkpoint kinases including ATM, ATR and Check2 are enriched in erythroid precursors. It is clear evidence against the common belief that erythroblasts would not need to 'invest' in DNA repair because of impending enucleation.

We share the reviewer's interest in the apparent DNA checkpoint activation in erythroid precursors. We edited the paragraph (Page 12, Line 364-373) to discuss potential causes of checkpoint activation by replicative stress and increased DNA damage-inducing ROS that occurs during elevated heme and hemoglobin metabolism.

4. The authors conclude that both Kit and MAPK pathways are downregulated early, whereas EpoR and Stat5 persist into later differentiation stages. They then draw the conclusion that Kit signals primarily through MAPK, whereas EpoR signals through Stat5. This conclusion is not justified simply based on data that would be correlative at best. Further, EpoR signaling through MAPK is well documented. The authors interpret the data in Figure 6D to suggest that EpoR remains uniformly high but it too seems to decline immediately following its peak in ProE, a finding consistent with previous reports using radioligand binding. Finally, the time course of the various MAPKs is more complex than the simple decline seen with Kit.

We agree that we may have oversimplified the interpretation of our findings. We have introduced some changes in the text that reinforce the reviewer's concerns. For example: "The erythroid cytokine receptors c-Kit and EPOR have distinct roles in erythropoiesis, although their signaling pathways overlap considerably" (Page 15, Lines 462-463). We have noted mechanisms by which EPOR can stimulate MAPK/ERK signaling (Page 18 and 19, Lines 570-587) and cite previous studies that support the concept that c-Kit activates ERK more potently than EPOR (Page 18, Line 576).

We do believe that our data illustrate the general trend that c-Kit signaling declines earlier in erythropoiesis and preferentially promotes MAPK/ERK signaling compared to EPOR signaling, although we agree that some of our wording could be more accurate. We modified the text accordingly. For example, we changed the text from "Compared to c-Kit, EPOR/JAK2 levels were stable until the poly stage" to "Compared to c-Kit, the expression of EPOR/JAK2 declined more slowly during erythropoiesis and was sustained to later stages" (Page 15, Lines 477-478).

16th Oct 2020

Manuscript Number: MSB-20-9813R, Integrative proteomics reveals principles of dynamic phospho-signaling networks in erythropoiesis

Thank you for sending us your revised manuscript. We think that the performed revisions satisfactorily address the issues raised by the reviewers. I am glad to inform you that we can soon accept your manuscript for publication, pending some editorial issues listed below.

2nd Authors' Response to Reviewers**23rd Oct 2020**

The authors have made all requested editorial changes

Accepted**26th Oct 2020**

Thank you again for sending us your revised manuscript. We are now satisfied with the modifications made and I am pleased to inform you that your paper has been accepted for publication.

Corresponding Author Name: Prof. Dr. Matthias Mann

Manuscript Number: MSB-20-9813R